# Caveolae internalization repairs wounded cells and muscle fibers

Matthias Corrotte[1†], Patricia E Almeida[1,2†], Christina Tam[1], Thiago Castro-Gomes[1], Maria Cecilia Fernandes[1], Bryan A Millis[3], Mauro Cortez[1], Heather Miller[1], Wenxia Song[1], Timothy K Maugel[4], Norma W Andrews[1*]

[1]Department of Cell Biology and Molecular Genetics, University of Maryland, College Park, United States; [2]CAPES Foundation, Brazil Ministry of Education, Brasilia, Brazil; [3]Laboratory of Cell Structure and Dynamics, National Institute on Deafness and Other Communication Disorders, National Institutes of Health, Bethesda, United States; [4]Laboratory for Biological Ultrastructure, University of Maryland, College Park, United States

**Abstract** Rapid repair of plasma membrane wounds is critical for cellular survival. Muscle fibers are particularly susceptible to injury, and defective sarcolemma resealing causes muscular dystrophy. Caveolae accumulate in dystrophic muscle fibers and caveolin and cavin mutations cause muscle pathology, but the underlying mechanism is unknown. Here we show that muscle fibers and other cell types repair membrane wounds by a mechanism involving $Ca^{2+}$-triggered exocytosis of lysosomes, release of acid sphingomyelinase, and rapid lesion removal by caveolar endocytosis. Wounding or exposure to sphingomyelinase triggered endocytosis and intracellular accumulation of caveolar vesicles, which gradually merged into larger compartments. The pore-forming toxin SLO was directly visualized entering cells within caveolar vesicles, and depletion of caveolin inhibited plasma membrane resealing. Our findings directly link lesion removal by caveolar endocytosis to the maintenance of plasma membrane and muscle fiber integrity, providing a mechanistic explanation for the muscle pathology associated with mutations in caveolae proteins.

*For correspondence: andrewsn@umd.edu

†These authors contributed equally to this work

Competing interests: The authors declare that no competing interests exist.

## Introduction

$Ca^{2+}$ entry in wounded cells triggers a repair mechanism that reseals the plasma membrane (PM) within a few seconds (*McNeil et al., 2003*). $Ca^{2+}$ influx induces exocytosis of lysosomes, a process required for PM resealing (*Reddy et al., 2001*). PM repair was initially suggested to be mediated by a membrane patch applied to the wound site (*Miyake and McNeil, 1995*), or through exocytosis-mediated reduction in PM tension (*Togo et al., 2000*). However, $Ca^{2+}$-dependent lysosomal exocytosis is also required for the resealing of cells injured by pore-forming toxins (*Walev et al., 2001*; *Idone et al., 2008*). These toxins generate stable, protein-lined transmembrane lesions that cannot be resealed by a membrane patch or simply by relieving PM tension. Recent studies clarified this issue, by showing that lysosomal exocytosis in wounded cells is followed by a rapid, cholesterol-dependent form of endocytosis that removes pores and lesions from the PM (*Idone et al., 2008*) and directs them to lysosomes for degradation (*Corrotte et al., 2012*).

$Ca^{2+}$-triggered exocytosis of the lysosomal enzyme acid shingomyelinase (ASM) is required for the endocytic process that promotes wounded cell resealing. Transcriptional silencing of ASM abolished PM repair, and addition of exogenous ASM restored resealing (*Tam et al., 2010*). These findings provided a novel conceptual framework for the mechanism of PM repair, indicating that exocytosis promotes resealing not by generating a membrane patch, but by releasing enzymes that remodel the outer leaflet of the PM and stimulate endocytosis. ASM converts the abundant PM lipid sphingomyelin

**eLife digest** Cells must be able to rapidly repair damage to their outer membranes. This is particularly important in the case of muscle cells, which are vulnerable to damage, and the failure of these cells to repair their outer membranes leads to the muscle wastage seen in muscular dystrophy. Researchers do not fully understand how cells repair membrane, but one popular theory is that they use the membranes of specialized vesicles to 'patch' areas that have been damaged.

A group of proteins called caveolins have also been implicated in membrane repair but, again, the details have not been worked out. These proteins are best known for their role in the formation of caveolae — small pouches formed by invaginated sections of the plasma membrane. Now, Corrotte et al. have obtained evidence that membrane repair relies not on patching, but on endocytosis (the process by which substances are taken into the cell in small vesicles that 'pinch' from the plasma membrane) of these caveolae pouches.

Corrotte et al. treated cells with streptolysin O, a toxin that forms pores in the membrane that cannot be repaired using patches, and found that this led to the formation of small membrane-derived vesicles that looked just like caveolae. Further tests confirmed that these vesicles contained caveolar proteins, and that they removed the toxin from the plasma membrane by endocytosis. Similar effects were seen in response to mechanical damage caused by tiny glass beads. Moreover, blocking the expression of caveolar genes prevented cells from repairing membrane damage.

Based on their findings, Corrotte et al. propose an alternative model for the repair process; namely that cellular damage triggers an influx of calcium ions, which causes vesicles called lysosomes to release chemicals that promote the formation of caveolae. These then remove the damaged area through endocytosis, restoring the integrity of the membrane. The results offer new insights into why mutations in caveolar proteins are associated with muscle disorders, including muscular dystrophy and cardiac dysfunction.

into ceramide (*Schissel et al., 1996*), and ceramide-enriched microdomains can trigger invagination of lipid bilayers (*Holopainen et al., 2000*; *Trajkovic et al., 2008*). However, the exact role of ceramide and the nature of the endocytic vesicles triggered by cell wounding are still unknown.

PM repair is of paramount importance in muscle. Muscle fibers are frequently injured in vivo (*McNeil and Khakee, 1992*) and failure to repair the sarcolemma causes muscular dystrophy (*Bansal and Campbell, 2004*). Intriguingly, the fragile muscle fibers from patients with Duchenne muscular dystrophy contain elevated numbers of caveolae-like vesicles (*Bonilla et al., 1981*; *Repetto et al., 1999*), and mutations in the muscle-specific caveolar protein caveolin-3 (Cav3) cause multiple forms of muscle pathology (*Gazzerro et al., 2010*). These observations, taken together with the recently uncovered role for endocytosis in PM repair (*Idone et al., 2008*), raised the possibility that caveolae-derived endocytic vesicles might play a direct, heretofore unrecognized role in the mechanism responsible for resealing PM wounds. In this study we show that injury-induced internalization of caveolae-derived vesicles is a dynamic process essential for the restoration of PM integrity.

## Results

### Permeabilization with a pore-forming toxin or exposure to sphingomyelinase induces accumulation of ceramide and <80 nm vesicles

Prior studies suggested that ASM released from lysosomes during cell wounding triggers formation of ceramide-enriched endocytic vesicles (*Tam et al., 2010*). By performing cryo-immuno EM assays with specific anti-ceramide antibodies (*Fernandes et al., 2011*) we detected anti-ceramide reactivity throughout the cytoplasm and in small clusters near the PM of NRK cells (*Figure 1A,B*, control). An isotype control antibody showed little, if any, labeling in the same preparations (not shown). Treatment with purified *Bacillus cereus* sphingomyelinase (SM) for 30 s enhanced the anti-ceramide staining along the PM. Permeabilization with the pore-forming toxin streptolysin O (SLO) had a similar effect, rapidly increasing the anti-ceramide reactivity at the cell periphery (*Figure 1A,B*). These results suggested that injury with SLO or exposure to SM triggered the formation of ceramide-enriched structures that might represent PM invaginations or intracellular vesicles.

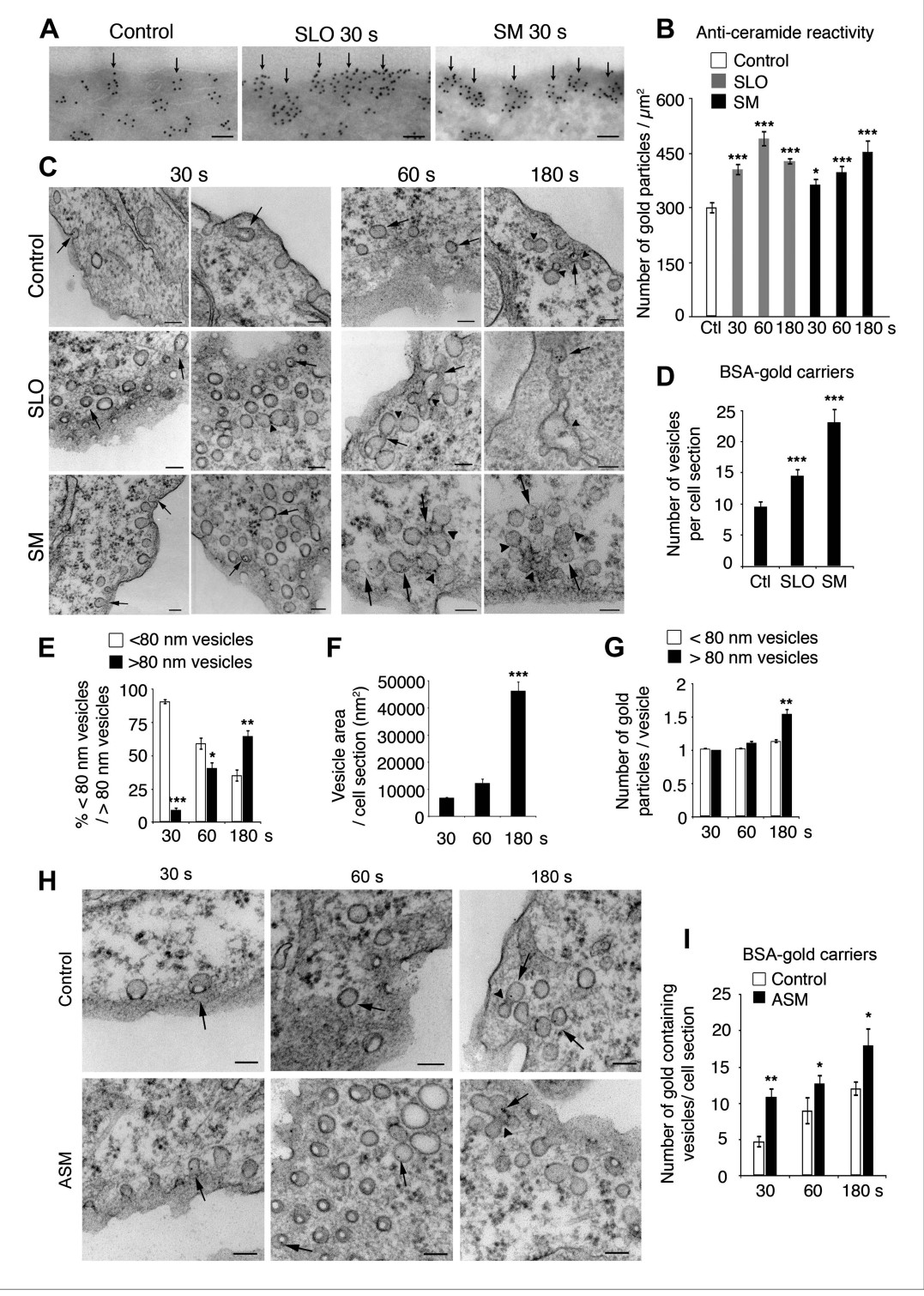

**Figure 1**. Caveolae-like vesicles accumulate in cells exposed to SLO and sphingomyelinase. (**A**) Cryo-immuno EM with anti-ceramide in NRK cells untreated or exposed to SLO or SM for 30 s. Bars: 100 nm. Arrows: patches of ceramide staining near the PM. (**B**) Quantification of anti-ceramide label in cells treated as in (**A**). All gold particles (2522–6876) within an area of 200 nm along the PM were counted in 14–31 cell sections. Data represent mean ± SEM of gold particles/cell section. *p=0.023, ***p<0.001. The results are representative of two independent experiments. (**C**) TEM of NRK cells exposed or not to SLO+Ca²⁺ or SM in the presence of BSA-gold. Arrows: <80 nm vesicles with BSA-gold. Arrowheads: merged vesicles. Bars: 100 nm. (**D**) Quantification of vesicles with BSA-gold in control,
*Figure 1. Continued on next page*

*Figure 1. Continued*

SLO or SM-treated cells after 30 s. All vesicles containing BSA-gold (191–485) were counted in 20 cell sections/sample. Data represent mean ± SEM of BSA-gold-containing vesicles/cell section. ***p<0.001. The results are representative of two independent experiments. (**E**) Numbers of BSA-gold positive <80 nm and >80 nm vesicles over time in SLO treated cells. Data represent mean ± SEM of vesicles/cell section. *p=0.033, **p=0.004, ***p<0.001 (comparison with <80 nm vesicles in the same time point). (**F**) Average area of BSA-gold positive vesicles over time. Data represent mean ± SEM of vesicle area/cell section. ***p<0.001 (comparison with 30 s time point). (**G**) BSA-gold particles detected within <80 nm and >80 nm vesicles over time. Data represent mean ± SEM of gold particles. **p=0.0019 (comparison with <80 nm vesicles in the same time point). From (**E**) to (**G**), all gold-containing vesicles (73–142) were quantified in 14–47 cell sections. (**H**) TEM of NRK cells untreated (control) or treated with ASM in the presence of BSA-gold as an endocytic tracer. Arrows point to <80 nm vesicles containing BSA-gold; arrowheads point to vesicle fusion profiles. Bars: 100 nm. (**I**) Quantification of BSA-gold containing vesicles over time in cells treated or not with ASM. All BSA-gold carriers (58–309) were counted in 10–20 sections. Data represent mean ± SEM of BSA-gold-containing vesicles/cell section. *p=0.03–0.04, **p=0.005 (comparison with controls in each time point). All datasets were compared using an unpaired Student's *t* test.

The following figure supplements are available for figure 1:

**Figure supplement 1**. Transcriptional silencing of ASM inhibits intracellular accumulation of caveolae-like vesicles after SLO injury.

To directly visualize newly formed structures, we examined cells by transmission electron microscopy (TEM) at increasing periods after permeabilization with SLO or exposure to SM. Previous TEM studies detected numerous large, irregularly shaped endocytic vesicles in cells fixed 4–5 min after SLO permeabilization (*Idone et al., 2008*). Surprisingly, when cells were examined just 30 s after treatment with SLO or SM, the newly formed endocytic vesicles (identified by luminal BSA-gold added as an endocytic tracer) appeared as homogeneously round and small (<80 nm). Similar peripheral <80 nm endocytic vesicles were present in untreated cells, albeit in lower numbers (*Figure 1C*). Quantification revealed that treatment with SLO or SM for 30 s increased the number of BSA-gold-containing vesicles relative to controls (*Figure 1D*). Clathrin-coated vesicles in the same preparations did not contain BSA-gold, in agreement with the slower rate of formation of this class of endocytic vesicles (results not shown). At later time points (60 and 180 s) larger compartments suggestive of homotypic fusion of the <80 nm vesicles were increasingly observed (*Figure 1C*). Quantification of vesicle size, area and BSA-gold content supported the conclusion that the small endocytic vesicles induced by exposure to SLO or SM increase in size over time (*Figure 1E–G*).

Notably, the number of <80 nm vesicles containing the endocytic tracer BSA-gold also increased when cells were treated with recombinant human ASM (*He et al., 1999*) (*Figure 1H,I*). Furthermore, transcriptional silencing of ASM reduced the number of peripheral <80 nm vesicles seen by TEM in cells exposed to SLO+Ca$^{2+}$ (*Figure 1—figure supplement 1*). These results reinforce the view that ASM released through lysosomal exocytosis in wounded cells can generate ceramide on the outer leaflet of the PM (*Schissel et al., 1998*) and promote endocytosis (*Tam et al., 2010*).

## SLO is removed from the PM by caveolar endocytosis

The newly-formed endocytic vesicles observed in SLO or SM-treated cells strongly resembled caveolae, the flask-like PM invaginations enriched in cholesterol and sphingolipids that are present in many cell types (*Palade, 1953*; *Parton and Simons, 2007*). To investigate a potential role of caveolae-derived vesicles in the internalization of SLO pores, cells were permeabilized with GFP-tagged SLO (which retains full pore-forming activity [*Idone et al., 2008*]) and analyzed by cryo-immuno EM using antibodies against GFP or the caveolae-associated protein caveolin-1 (Cav1) (*Drab et al., 2001*). The amount of GFP-SLO associated with flat regions of the PM gradually decreased over time, consistent with a toxin internalization process (*Figure 2A,B*). Importantly, during the first 60 s after injury GFP-SLO was increasingly detected on <80 nm vesicles containing Cav1, which are properties of caveolae (*Figure 2A,C*). By 300 s the amount of SLO co-localizing with Cav1 decreased, in agreement with the previously described traffic of internalized SLO into later compartments of the endocytic pathway (*Corrotte et al., 2012*). The number of <80 nm vesicles positive for Cav1 alone or SLO alone also decreased over time, simultaneously with an increase in the number of >80 nm vesicles containing either Cav1 alone or both Cav1 and SLO (*Figure 2C,D*). These results are fully consistent with our TEM

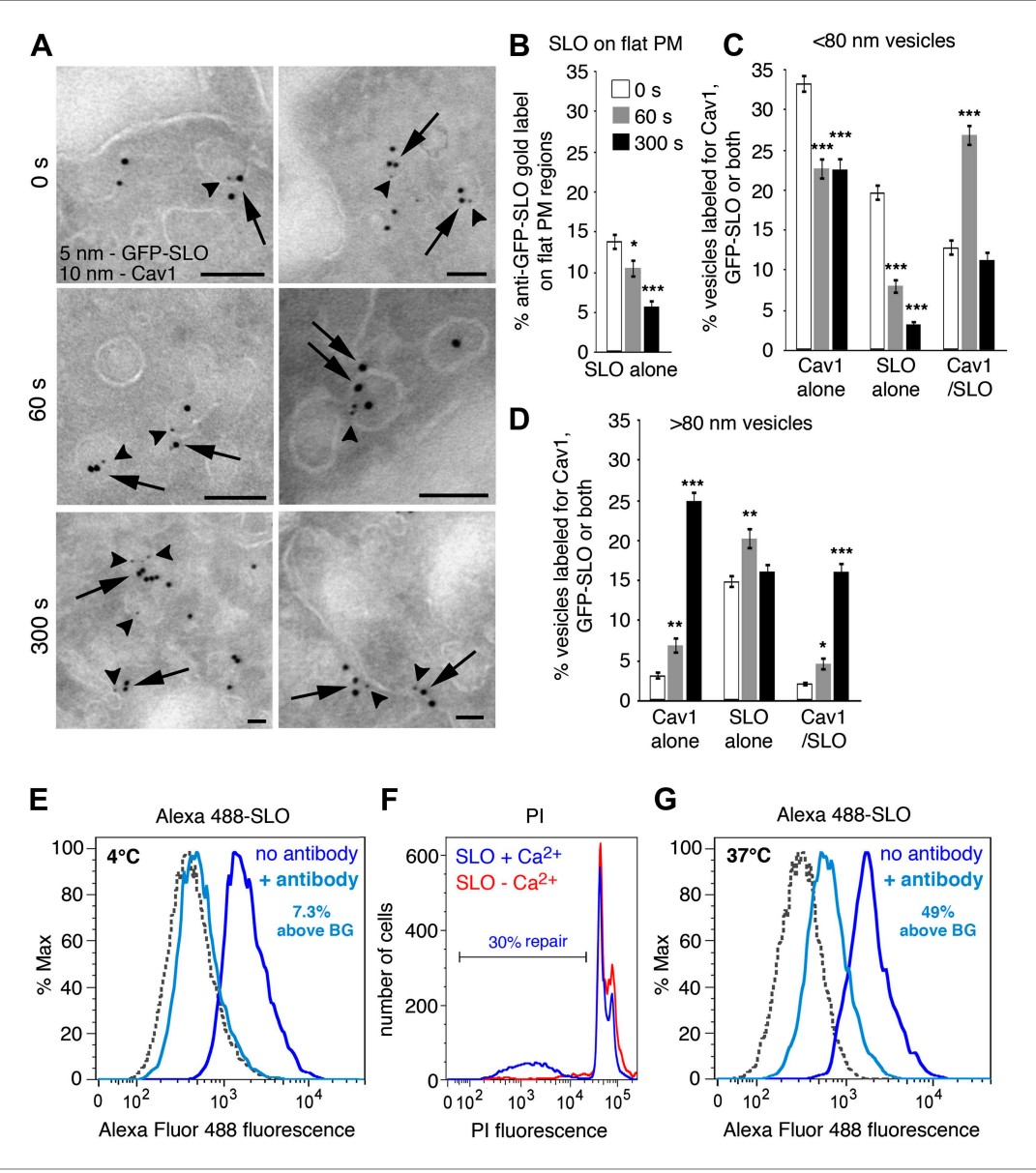

**Figure 2**. SLO is internalized in Cav1-positive caveolae-like vesicles that separate from the PM. (**A**) Cryo-immuno EM localization of GFP-SLO and Cav1 in NRK cells. 5 nm gold: anti-GFP (arrowheads); 10 nm gold: anti-Cav1 (arrows). Bars: 100 nm. (**B–D**) Quantification of the relative amount of GFP-SLO and/or Cav1 on flat PM structures (**B**), vesicular profiles <80 nm (**C**) or vesicular profiles >80 nm (**D**). All labeled structures (17–280) in 80 random fields were counted and the data expressed as % of total antibody-stained structures. Data represent mean ± SEM of labeled structures/ cell section. *p=0.039–0.052, **p=0.006–0.007, ***p<0.001, unpaired Student's $t$ test. The results are representative of two independent experiments. (**E**) FACS analysis of NRK cells exposed to Alexa 488-SLO at 4°C for 5 min, followed by anti-Alexa Fluor 488 quenching antibodies for 2 min. The percentage of quench-protected toxin fluorescence above the endogenous cellular background level (BG) is indicated. (**F**) FACS analysis of NRK cells exposed to Alexa 488-SLO ± Ca²⁺ at 37°C for 5 min, followed by PI staining. The percentage of PI-negative cells is indicated. (**G**) FACS analysis of NRK cells exposed to Alexa 488-SLO + Ca²⁺ at 37°C for 5 min, followed by anti-Alexa Fluor 488 quenching antibodies for 2 min. The profile shown corresponds to the PI-negative cell population shown in (**F**). The percentage of quench-protected toxin fluorescence above the endogenous cellular background level (BG) is indicated. Dashed lines, no Alexa 488-SLO background controls. The results are representative of at least five independent experiments.

analysis of SLO-permeabilized cells, showing an initial increase in the number of <80 nm endocytic vesicles followed by the gradual appearance of merged compartments (*Figure 1C*).

The results described above strongly suggested that SLO pores were removed from the PM by internalization in Cav1-positive caveolar vesicles. However, the possibility still remained that caveolar vesicles containing SLO might still be connected to the PM. To directly address this issue, we developed a flow cytometry (FACS) assay for detection of cell-associated fluorescent SLO before and after quenching with specific antibodies. To ensure accessibility of the fluorescent moiety to the quenching antibodies, we used a single cysteine SLO mutant labeled with Alexa Fluor 488 at the N-terminus (a region not inserted into membranes during pore-formation [*Shatursky et al., 1999*]). When cells were incubated with the labeled toxin at 4°C, addition of anti-Alexa Fluor 488 antibodies quenched >90% of the fluorescence (*Figure 2E*). Thus, under conditions that allow toxin binding but not endocytosis, most cell-associated toxin is accessible to quenching antibodies. When the same amount of labeled toxin was added to cells at 37°C, cells were fully permeabilized and about 30% resealed in the presence of $Ca^{2+}$ under the assay conditions, as indicated by propidium iodide (PI) exclusion (*Figure 2F*). By gating on the PI-negative (resealed) cell population we found that at least 50% of the cell-associated Alexa 488-SLO was protected from quenching (*Figure 2G*), indicating that it entered compartments no longer in contact with the extracellular medium. Antibody-mediated quenching of extracellular Alexa 488-SLO was also observed by confocal microscopy (*Figure 3A*), and quench-protected intracellular toxin colocalized with Cav1-positive puncta (*Figure 3B*).

To obtain dynamic data on the behavior of SLO carriers, we performed live imaging of cells expressing mRFP-Cav1 and permeabilized with GFP-SLO. The transfected mRFP-Cav1 was detected on peripheral punctate structures, in a pattern indistinguishable from the distribution of endogenous Cav1 (*Figure 4A*). A few seconds after pore formation was triggered by increasing the temperature, GFP-SLO was observed moving into cells in structures containing mRFP-Cav1 (*Figure 4B*, arrowheads; *Videos 1, 2, 3*). Some Cav1-positive SLO carriers merged intracellularly while moving deeper into cells (*Figure 4B* arrows; *Video 3*). Kymograph analysis detected numerous Cav1-positive SLO carriers entering a peripheral intracellular region of SLO-permeabilized cells (*Figure 4C*, lower panels). In contrast, only a few intracellular Cav1-containing structures were detected at the periphery of untreated cells (*Figure 4C*, upper panels). While some GFP-SLO signal was present in compartments with no detectable Cav1, the majority of SLO carriers detected in the kymographs co-localized with the caveolae marker (*Figure 4C*, lower panels).

The ATPase EHD2 recently emerged as a regulator of caveolae dynamics. By linking caveolae to actin filaments, EHD2 was proposed to retain caveolae on the PM, preventing their internalization (*Moren et al., 2012*; *Stoeber et al., 2012*). As an additional independent assay for detecting SLO-induced caveolar endocytosis, we examined the co-localization of EHD2 with Cav1 clusters at the PM by TIRF microscopy, after cells were permeabilized with SLO or treated with SM. Most of the Cav1 staining at the surface of untreated cells appeared as puncta that were also positive for EHD2, as previously described for the stable population of PM-associated caveolae (*Stoeber et al., 2012*). The intensity of Cav1 puncta decreased with time after SLO or SM exposure, reflecting an inward movement away from the bright TIRF field adjacent to the PM (*Figure 5A,B*). In addition, cells permeabilized with SLO or exposed to SM showed a progressive loss in Cav1-EHD2 co-localization (*Figure 5A,C*), as expected for caveolar endocytosis.

Collectively, our assays showing co-localization of endogenous Cav1 and GFP-SLO on <80 nm vesicles (*Figure 2A–D*), detection of internalized Alexa 488-SLO after extracellular quenching (*Figure 2E–G and 3*), live imaging of GFP-SLO entering cells in vesicles containing mRFP-Cav1 (*Figure 4B,C* and *Videos 1–3*) and loss of Cav1 and EHD2 co-localization after exposure to SLO (*Figure 5A–C*) support the conclusion that the pore-forming toxin SLO is removed from the PM in Cav1-positive, caveolae-derived endocytic vesicles. Furthermore, all assays also indicate that SLO internalization occurs within a few seconds of pore formation, coinciding with the known kinetics of PM resealing (*Steinhardt et al., 1994*; *Idone et al., 2008*).

## Cav1 is required for internalization of caveolar vesicles and PM repair

Cav1 is required for caveolae assembly in many cell types (*Drab et al., 2001*; *Parton and Simons, 2007*). To examine the requirement for caveolae in PM repair, we transcriptionally silenced Cav1 expression (*Figure 6A*) and examined the ability of cells to reseal after SLO injury using a live imaging assay that follows the influx of the lipophilic dye FM1–43 (*Idone et al., 2008*; *Tam et al., 2010*).

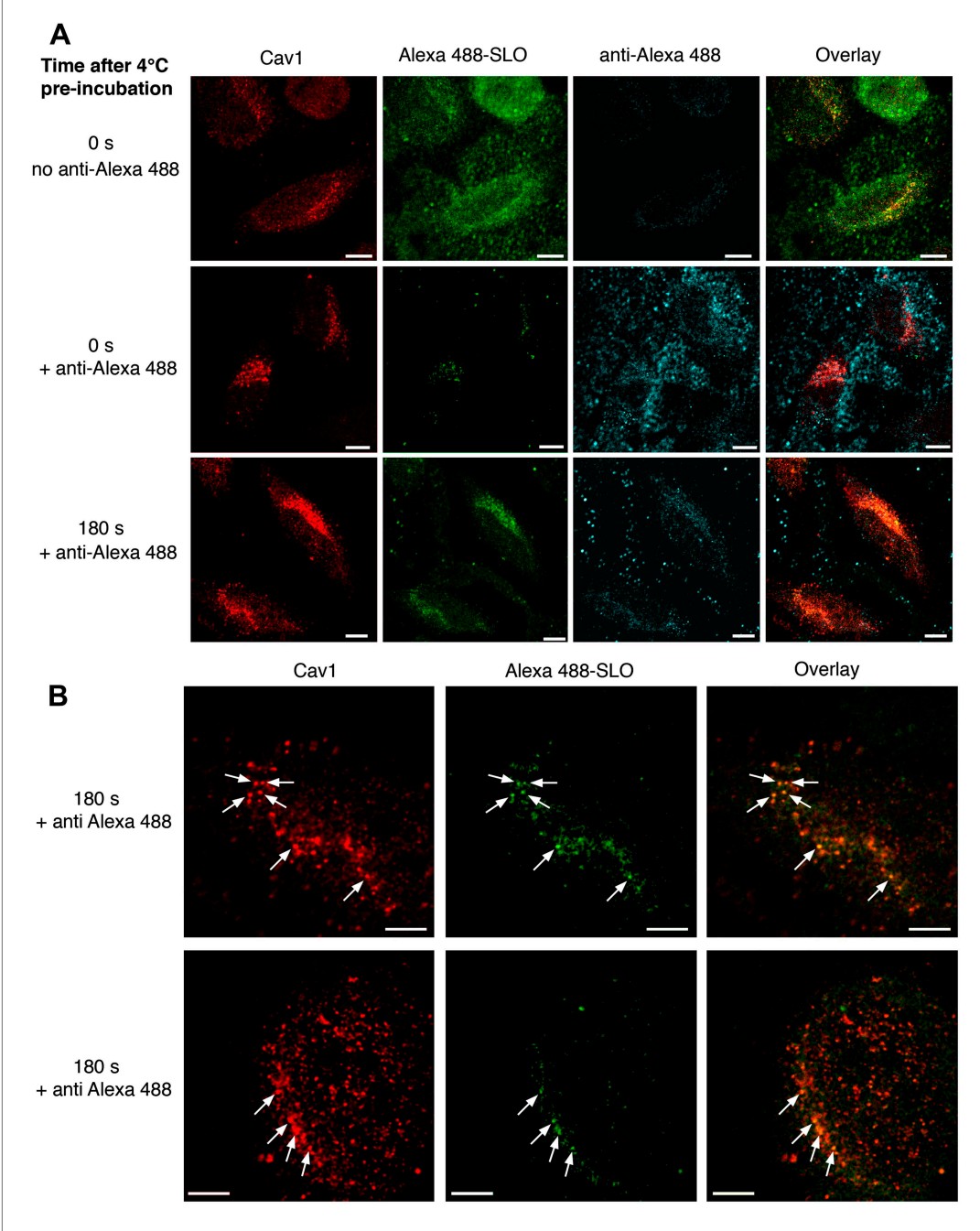

**Figure 3**. Internalized SLO colocalizes with Cav1. (**A**) HeLa cells were pre-incubated with 3 µg/ml Alexa 488-SLO (green) for 5 min at 4°C, washed and either kept at 4°C (0 s) or incubated at 37°C in DME+Ca²⁺ (180 s), followed or not by anti-Alexa Fluor 488 quenching antibodies (blue) for 30 min at 4°C. After fixation cells were permeabilized, labeled with anti-Cav1 antibodies (red) and analyzed by confocal microscopy (single optical sections are shown). Bars: 10 µm. (**B**) Higher magnification images of cells treated as in **A** and incubated for 180 s at 37°C with quenching antibodies. Arrows: Vesicular carriers positive for both Cav1 and Alexa 488-SLO. Bars: 5 µm.

FM1–43 was detected only on the PM of cells not exposed to SLO, consistent with an intact lipid bilayer. After treatment with SLO in the absence of Ca²⁺ (a condition that does not allow PM repair) there was massive FM1–43 influx, reflecting rapid PM permeabilization (***Figure 6B***, ***Video 4***). Quantification of FM1–43 influx showed that cells treated with control or Cav1 siRNA were similarly susceptible to SLO permeabilization (***Figure 6C***). In the presence of Ca²⁺, SLO did not trigger FM1–43

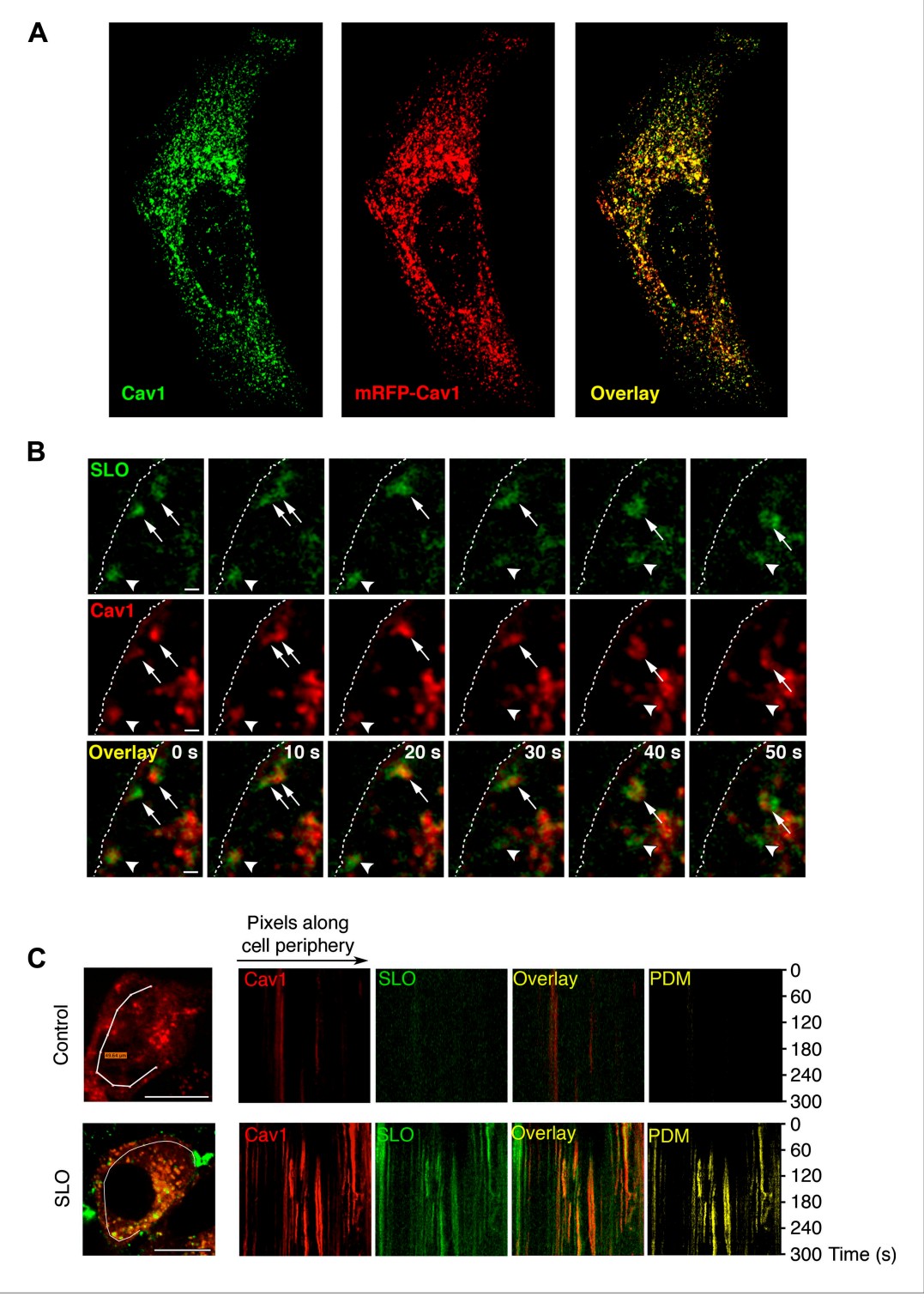

**Figure 4**. SLO enters cells associated with Cav1-positive carriers. (**A**) Confocal optical section at the bottom surface of a HeLa cell transfected with mRFP-Cav1 (red) and stained with anti-Cav1 antibodies (green). (**B**) Time-lapse images of HeLa cells expressing mRFP-Cav1 and imaged for 300 s after incubation with GFP-SLO+$Ca^{2+}$. Dotted line: PM. Arrowheads: Cav1/SLO carrier moving rapidly into cell. Arrows: Cav1/SLO carriers that merge after internalization. Bars: 1 μm. See also *Figure 2* and *Videos 1, 2, 3*. (**C**) Kymographs of mRFP-Cav1 and GFP-SLO fluorescence along a line at the cell periphery. PDM: Positive Difference of the Mean (co-localization index). Bars: 10 μm. The results are representative of seven independent experiments.

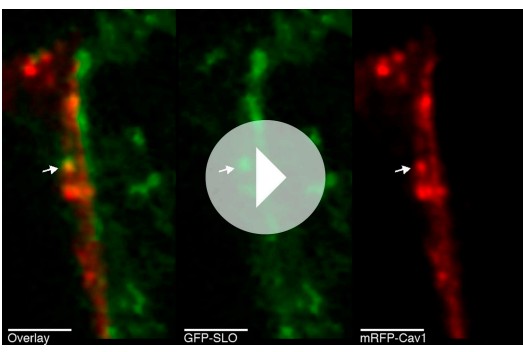

**Video 1**. Internalization and lateral movement of SLO/Cav1 carriers, (related to *Figure 4*). HeLa cells expressing mRFP-Cav1 (red) were pre-incubated for 5 min with 800 ng/ml of GFP-SLO (green) at 4°C and transferred in cold DMEM+Ca²⁺ to a live imaging chamber at 37°C, to allow for progressive warming, pore formation, and PM repair. Images were acquired for 5 min at 2 s/frame on a spinning disk confocal microscope. The video shows a vesicular structure positive for Cav1 and SLO that appears to separate rapidly from the PM and move laterally along the PM before disappearing into a different focal plane. Video is displayed at 6.67 frames/s. Dotted line: PM. Arrow indicates Cav1/SLO carrier. Bars: 5 μm.

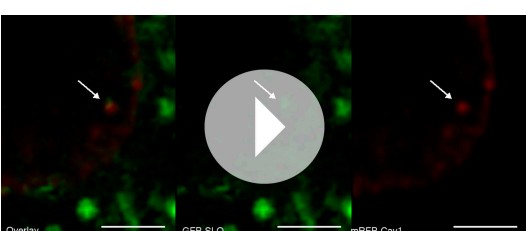

**Video 2**. Internalization and trafficking of SLO/Cav1 carriers (related to *Figure 4*). HeLa cells were treated as in *video 1*. The video shows a Cav1 positive structure on the PM that accumulates SLO before being internalized and moving rapidly into the cell. Video is displayed at 6.67 frames/s. Dotted line: PM. Arrow indicates Cav1/SLO positive vesicle. Bars: 5 μm.

influx in cells treated with control siRNA, as expected from the rapid Ca²⁺-dependent resealing process (*Idone et al., 2008*; *Tam et al., 2010*). In contrast, FM1–43 flowed rapidly into cells treated with Cav1 siRNA even in the presence of Ca²⁺, reflecting defective PM repair (*Figure 6B,C*).

All <80 nm vesicles with caveolae-like morphology observed along the cell periphery were quantified by TEM, and the results indicated that SLO or SM exposure increases the number of caveolae-like vesicles in control cells, but not after treatment with Cav1 siRNA (*Figure 6D,E*). Thus, injury with SLO or exposure to the ceramide-generating enzyme SM increase the population of peripheral caveolae-like vesicles in a Cav1-dependent manner, a process that correlates with the ability of these cells to reseal their PM. Quantification by cryo-immuno EM revealed that ~20% of the Cav1 detected on sections of NRK cells was initially associated with flat PM regions without caveolae, and this fraction was reduced to ~9% after 60 s of exposure to SLO+Ca²⁺, along with an increase in labeling of intracellular <80 nm vesicles (*Figure 6F*). These results suggest that the intracellular accumulation of caveolae-like vesicles after exposure to SLO or SM results from internalization of pre-existing and, to a certain extent, of de novo assembled caveolae.

Surprisingly, we found no requirement for dynamin-1 and -2 in the internalization of caveolar vesicles triggered by SLO permeabilization or SM exposure. Cells depleted in dynamin-2 with Dyn2 siRNA (*Figure 7A*) were strongly deficient in transferrin endocytosis (*Figure 7B*), but fully capable of repairing their PM after permeabilization with SLO+Ca²⁺ (*Figure 7C*). SLO-treated Dyn2-deficient cells contained numerous caveolae-like <80 nm vesicles that were not stained by externally added ruthenium red, consistent with a complete fission from the PM. Clathrin-coated vesicles in the same preparations were strongly stained with extracellular ruthenium red and showed elongated 'necks' in continuity with the PM, typical features of dynamin deficiency (*Figure 7D*). In sharp contrast, no similar structures

were observed associated with caveolae. To examine a potential compensatory role of dynamin-1, we performed similar assays using an inducible fibroblast cell line generated from dynamin-1 and -2 double conditional knockout mice (*Ferguson et al., 2009*). After tamoxifen induction these cells became strongly depleted in both dynamin-1 and dynamin-2 (*Figure 7E*) and defective in transferrin endocytosis (*Figure 7F*). However, after exposure to SLO+Ca²⁺ or SM these cells were still fully capable of blocking FM1–43 and propidium iodide (PI) influx (*Figure 7G*) and of upregulating endocytosis of the B subunit of cholera toxin (which is internalized by caveolar and other forms of endocytosis [*Kirkham et al., 2005*; *Chinnapen et al., 2007*]) (*Figure 7H*). These results suggest that caveolar endocytosis induced by PM injury or by exposure to the ceramide-generating enzyme SM may occur independently of dynamin function.

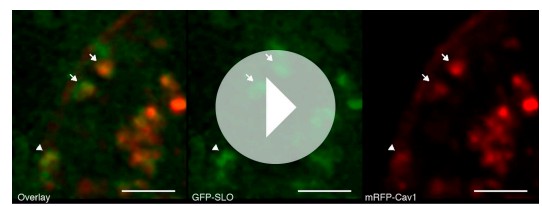

**Video 3**. Intracellular merger and rapid internalization of SLO/Cav1 carriers (related to *Figure 4*). HeLa cells were treated as in *video 1*. The video shows two separate SLO and Cav1 positive carriers close to the PM (arrows) that merge before rapidly moving further into the cell. The same cell contains a SLO/Cav1 positive structure (arrowhead) that suddenly separates from the PM and moves rapidly into the cell. Video is displayed at 6.67 frames/s. Dotted line: PM. Bars: 5 µm.

## Caveolae-like vesicles accumulate next to mechanical wounds and resealing is Cav1 dependent

We also assessed the involvement of caveolar endocytosis in the resealing of mechanical lesions. When cells were injured by exposure to glass beads, wounds were visualized by TEM as sites where ruthenium red entered the cytosol (*Figure 8A,B*, *Figure 8—figure supplement 1*, large arrows). 'Hot spots' of caveolae-like vesicle accumulation, often arranged in rows, were frequently seen close to wound sites. Several peripheral caveolae-like vesicles observed 30–60 s after wounding appeared disconnected from the PM, based on the lack of luminal staining with ruthenium red added during fixation (*Figure 8B*, *Figure 8—figure supplement 1A*, small arrows). At later time points, more complex structures suggesting homotypic fusion of caveolar vesicles were also observed, and some of these merged compartments were still connected to the PM, as indicated by luminal staining with ruthenium red (*Figure 8B*, 180 s).

Mechanical wounding with glass beads cannot be performed simultaneously with live imaging, so instead of monitoring FM1–43 influx we performed end-point PM repair assays based on exclusion of the membrane impermeable dye PI. As expected, in the absence of $Ca^{2+}$ wounds were not repaired, allowing injured cells to be identified by nuclear PI staining. In the presence of $Ca^{2+}$, a reduction in PI positive nuclei consistent with PM repair was observed in cells treated with control siRNA. In contrast, cells treated with Cav1 siRNA and injured in the presence of $Ca^{2+}$ showed significantly higher PI staining when compared to controls (*Figure 8C,D*). Similar results were obtained after cells were wounded by scraping from the dish, followed by PI staining and FACS quantification of the whole cell population (*Figure 8E*). Thus, repair of mechanical wounds on the PM is also inhibited after depletion of the caveolar protein Cav1.

## Injured muscle fibers secrete ASM and require Cav3 for sarcolemma repair

Mutations in the muscle-specific caveolin isoform Cav3 are associated with muscular dystrophy and other serious muscle abnormalities (*Gazzerro et al., 2010*). However, the role of Cav3 in muscle pathology is not fully understood, and was previously attributed to indirect effects not linked to caveolae formation and/or endocytosis (*Hernández-Deviez et al., 2008*; *Gazzerro et al., 2010*). We investigated this issue using C2C12 myoblasts, a cell line that faithfully reproduces muscle differentiation upon serum withdrawal, forming multi-nucleated myotubes that express muscle differentiation markers (*Blau et al., 1983*; *Kim et al., 2006*; *Figure 9A,B*). Both myoblast and myotube-enriched cultures were susceptible to permeabilization by SLO, and resealed in the presence of $Ca^{2+}$ (*Figure 9C*).

Myotubes responded to SLO permeabilization and $Ca^{2+}$ influx with exocytosis of lysosomes, as previously shown in other cell types (*Rodríguez et al., 1997*). A Lamp1 luminal epitope was detected on the sarcolemma surface (*Figure 9D*), and the lysosomal enzymes ß-hexosaminidase (data not shown) and ASM were released by myoblasts (undifferentiated culture, day 0) and myotubes (differentiated culture, day 4) (*Figure 9E*). Injury with SLO also enhanced endocytosis in muscle fibers, detected by using fluorescent dextran as a fluid phase tracer. Untreated myotubes displayed very few dextran-positive endocytic vesicles after 4 min, reflecting low levels of endocytosis during this time period. After exposure to SLO in the absence of $Ca^{2+}$ dextran filled the myotubes cytoplasm, reflecting permeabilization of the sarcolemma. In the presence of $Ca^{2+}$, a condition that allows PM repair, dextran was excluded from the myotube cytosol but was detected in a punctate intracellular pattern suggestive of endocytic vesicles (*Figure 9F*).

When untreated myotubes were examined by TEM, caveolae-like <80 nm invaginations were observed associated with the sarcolemma (*Figure 10A*, control). Similar to what we observed in other cell types, a 30 s treatment with SLO or SM markedly increased the number of <80 nm caveolae-like

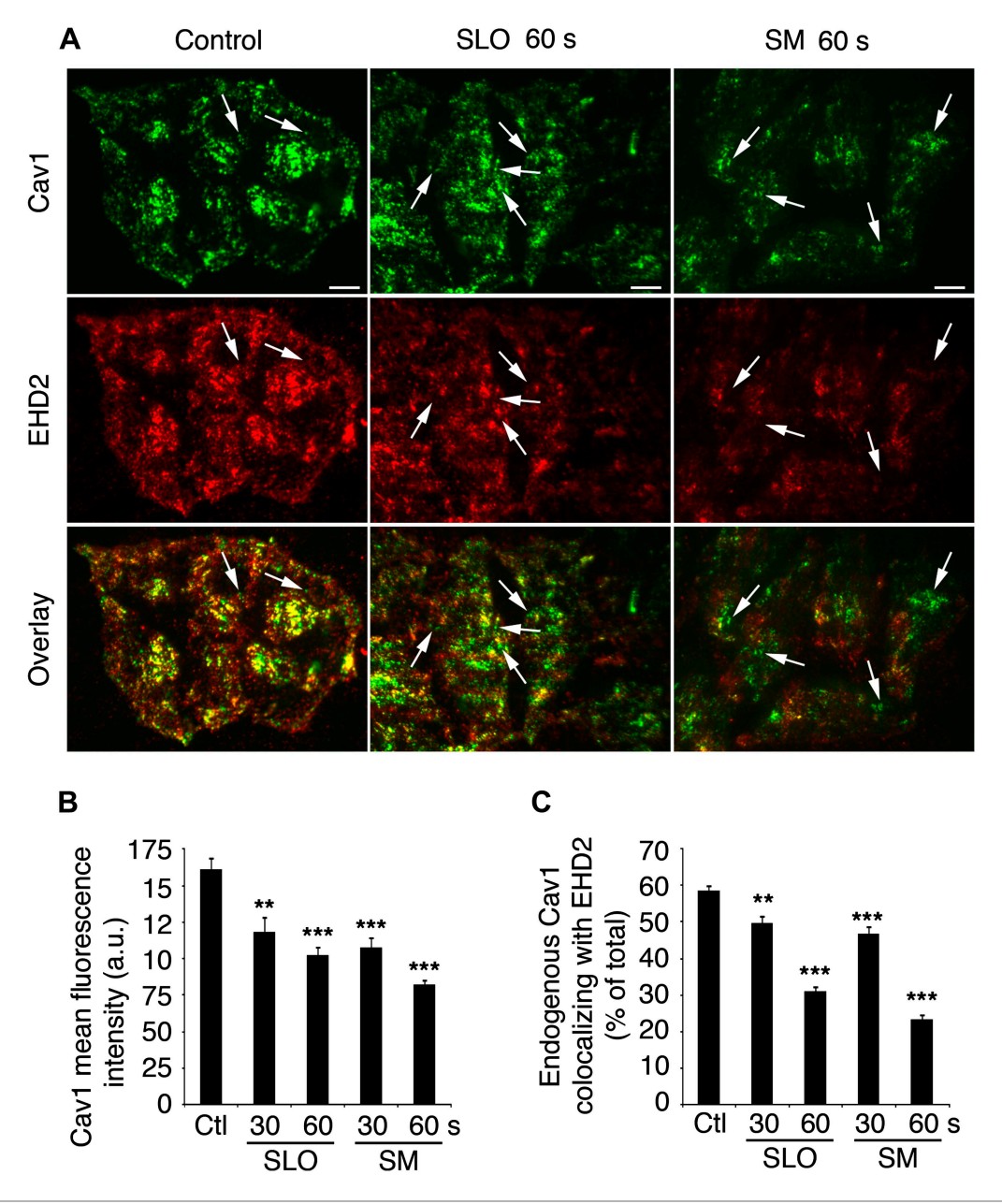

**Figure 5**. EHD2-Cav1 colocalization is decreased after caveolar endocytosis triggered by SLO or sphingomyelinase. (**A**) TIRF images of Cav1 and EHD2 immunostaining in HeLa cells treated or not with SLO or SM for 30 or 60 s. Bars : 50 μm. Arrows: Cav1 positive, EHD2 negative puncta corresponding to internalized caveolar vesicles. (**B**) Quantification of Cav1 fluorescence intensity in cells treated as in **C**. Quantifications were performed on 53–79 cells/sample. \*\*p=0.002, \*\*\*p<0.001, unpaired Student's *t* test. (**C**) Colocalization of Cav1 with EHD2 in cells treated as in **B**. Quantifications were performed on 65–82 cells/sample. Data represent mean ± SEM of values/cell. \*\*p=0.009, \*\*\*p<0.001, unpaired Student's *t* test. The results in **A**–**C** are representative of four independent experiments.

profiles observed in myotube sections (*Figure 10A*, SLO and SM). At later time points these profiles appeared larger and more complex, suggesting an ongoing process of vesicle homotypic fusion (*Figure 10A*, 180 s). Suggesting a certain degree of de novo caveolae assembly, immunofluorescence of myotubes exposed for 30 s to SM revealed an increase in the fraction of endogenous cavin1 colocalizing with Cav3 (*Figure 10B,C*).

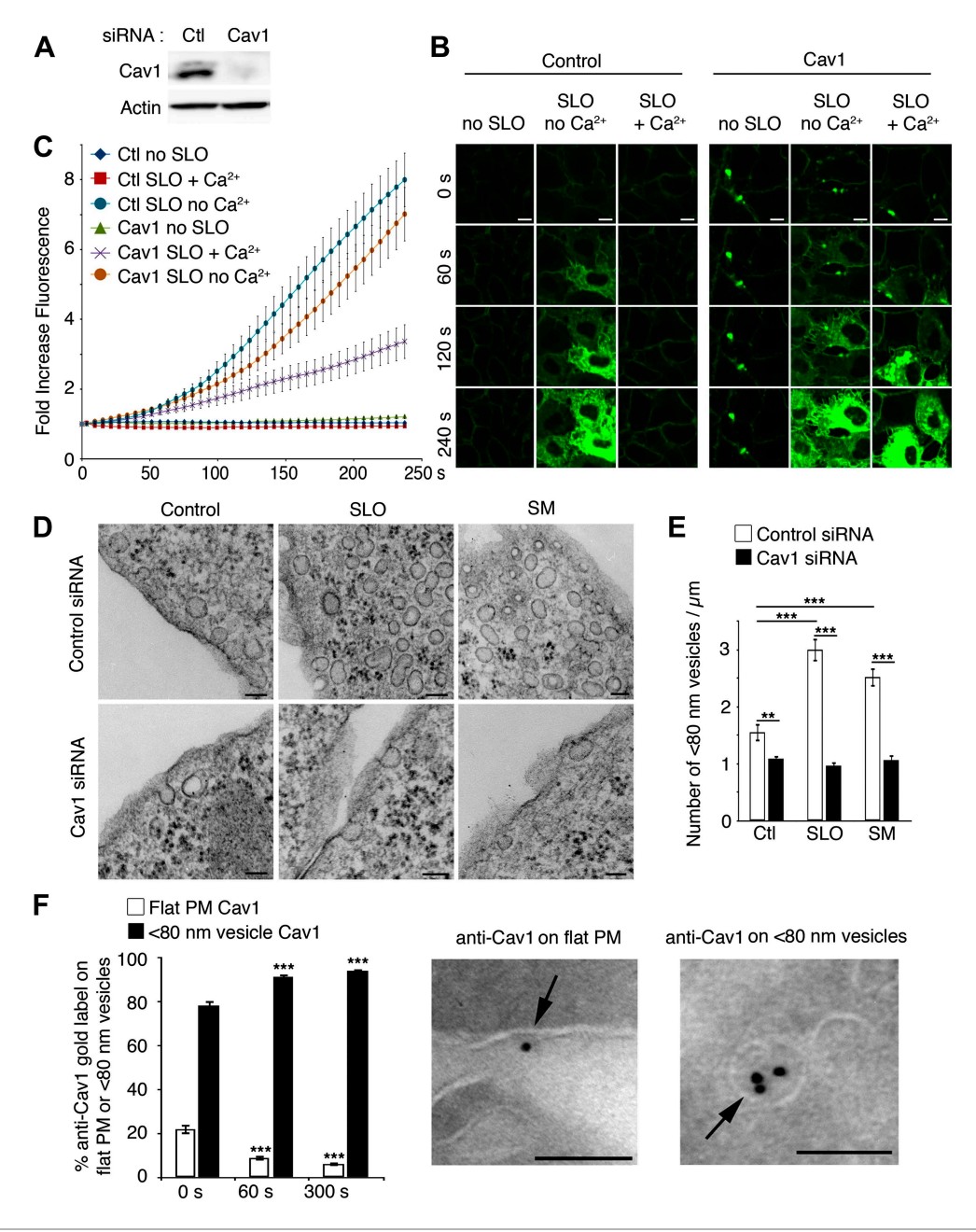

**Figure 6**. PM repair and endocytosis of SLO in caveolar vesicles are Cav1-dependent. (**A**) Western blot with anti-Cav1 and anti-actin (loading control) in NRK cell lysates treated with control or Cav1 siRNA. (**B**) Live imaging of FM1–43 influx in NRK cells treated with control or Cav1 siRNA, with and without SLO ± Ca$^{2+}$. Bars: 10 μm. See **Video 4.** (**C**) Quantification of intracellular FM1–43 fluorescence in **B**. Data represent mean ± SEM of fluorescence intensity/ cell. The results are representative of four independent experiments. (**D**) TEM of control and Cav1 siRNA-treated NRK cells incubated or not with SLO or SM for 30 s. Bars: 100 nm. (**E**) Number of <80 nm vesicular profiles/μm in cells treated as in **D**. All vesicles <80 nm diameter (1411–3912) were counted in 20–30 sections/sample and normalized by PM length. Data represent mean ± SEM of vesicles/cell section. **p=0.003, ***p<0.001, unpaired Student's $t$ test. The results are representative of two independent blinded quantifications performed by two independent investigators. (**F**) Cryo-immuno EM localization of Cav1 in NRK cells. All labeled structures (281–418) in 80 random fields were counted and the data expressed as % of total antibody stained structures. Data represent mean ± SEM of gold particles/ cell section. ***p<0.001 (comparison with 0 s condition), unpaired Student's $t$ test. The panels on the left show representative images of Cav1 staining (arrows, 10 nm gold label) on PM and <80 nm vesicular profiles. Bars: 100 nm. The results are representative of two independent experiments.

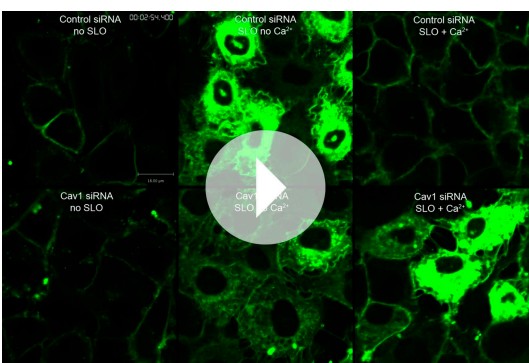

**Video 4**. RNAi-mediated silencing of Cav1 expression inhibits PM repair, allowing sustained FM1–43 influx into SLO-permeabilized NRK cells (related to **Figure 6**). NRK cells treated with control or Cav1 siRNA were left untreated (no SLO) or pre-incubated with SLO at 4°C and transferred to a live imaging chamber at 37°C in the presence or absence of Ca²⁺, followed by addition of FM1–43 and time-lapse imaging in a spinning disk confocal microscope for 4 min at 1 frame/3 s. Video is displayed at 10 frames/s. Bar: 18 µm.

The PM repair capacity of myotubes transcriptionally silenced for Cav3 expression was also reduced. Ca²⁺-free exposure to SLO resulted in PI staining of most myotube nuclei, reflecting the expected high levels of permeabilization and poor resealing. In the presence of Ca²⁺ very few PI positive nuclei were observed, demonstrating that myotubes effectively remove SLO pores from their sarcolemma in a Ca²⁺-dependent fashion. In contrast, Cav3-deficient myotubes exposed to SLO+Ca²⁺ showed high levels of nuclear PI staining, indicating a sarcolemma repair defect (**Figure 10D,E**). No nuclear staining with PI was detected in C2C12 myotube cultures not exposed to SLO, with or without Ca²⁺ (results not shown). These results actually underestimate the repair defect of Cav3-deficient myotubes, since C2C12 myotube cultures also contain undifferentiated myoblasts not expressing Cav3, which reseal their PM after SLO+Ca²⁺ permeabilization (**Figure 9B,C**). Thus, myotubes respond to wounding and Ca²⁺ influx with lysosomal exocytosis, secretion of ASM and intracellular accumulation of caveolar vesicles, a process that seems to be required for sarcolemma repair.

## Caveolae-like vesicles accumulate at sites of mechanical injury and at the periphery of primary muscle fibers after exposure to SLO and SM

Primary mouse *flexor digitorum brevis* muscle fibers were also examined after exposure to SLO or SM after 30 s. Without Ca²⁺, SLO triggered PI influx and staining of the fiber nuclei. With Ca²⁺, PI influx was blocked in most fibers, reflecting robust resealing. No nuclear PI staining was detected in fibers treated with SM, demonstrating that exposure to this enzyme does not impair sarcolemma integrity (**Figure 11**). As seen in C2C12 myotubes, treatment with SLO or SM resulted in increased numbers of intracellular caveolae-like vesicles along the sarcolemma, when compared to untreated controls (**Figure 12A–C**). Dissection resulted occasionally in localized fiber wounding (**Figure 11** small arrows, **Figure 12D**, **Figure 12—figure supplement 1**), providing an opportunity to examine the effect of mechanical wounding in primary muscle fibers. Intact regions along the fiber perimeter contained mostly a single layer of caveolae-like profiles close to the sarcolemma (**Figure 12D** panels 3 and 4—see whole data set in **Figure 12—figure supplement 1**). In contrast, an increased density of membrane profiles strongly resembling single and merged caveolae was evident in the proximity of wounds (**Figure 12D** panels 1 and 2, **Figure 12—figure supplement 1**). These observations are consistent with the view that primary muscle fibers, similarly to other cell types analyzed in this study, respond to injury with a rapid Ca²⁺-dependent resealing process that involves intracellular accumulation of caveolae-like vesicles.

## Discussion

The goal of this study was to investigate the abundant, cholesterol-dependent endocytic compartments that accumulate in several cell types after PM wounding (**Idone et al., 2008**; **Thiery et al., 2011**). Strikingly, our results revealed that a large fraction of the endocytic vesicles formed a few seconds after injury resemble caveolae, the ubiquitous flask-shaped PM invaginations that have been implicated in transcytosis, mechanosensing and signaling responses (**Parton and Simons, 2007**; **Lajoie and Nabi, 2010**). Wounded cells show increased numbers of small, homogeneously shaped vesicles with the typical morphology and markers of caveolae. Several independent lines of evidence demonstrate that internalization of these vesicles occurs rapidly, with a time-course that matches the kinetics of PM repair. Cryo-immuno EM, endocytosis and live imaging assays showed that the pore-forming toxin SLO is removed from the PM and traffics into cells in vesicular carriers containing the caveolar marker Cav1. Furthermore, transcriptional silencing of Cav1 in non-muscle cells and Cav3 in

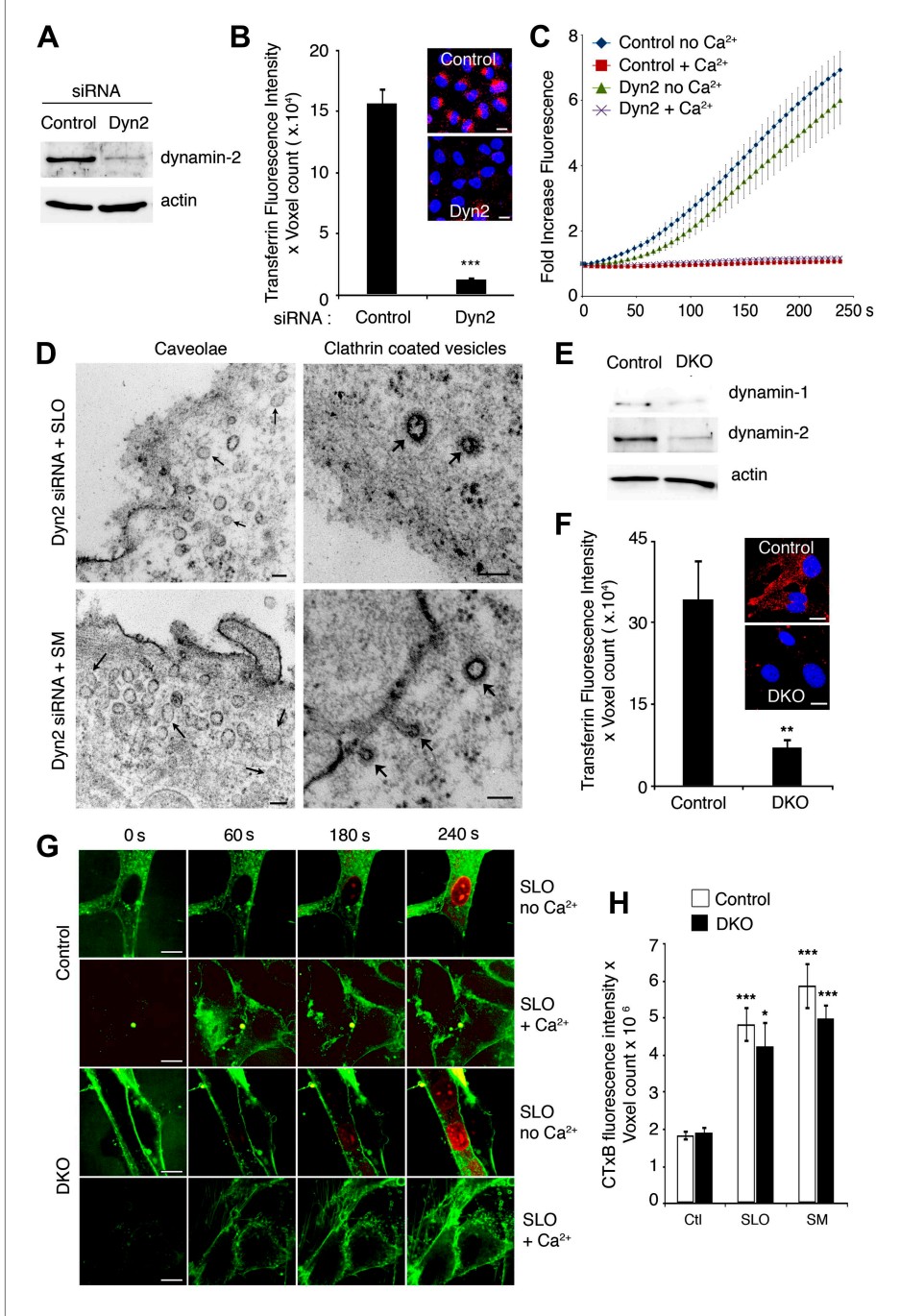

**Figure 7**. Depletion in dynamin-1 and -2 does not inhibit PM repair and SLO or SM-induced internalization of caveolar vesicles. (**A**) Western blot with anti-dynamin-2 and anti-actin (loading control) in lysates of NRK cells treated with control or Dyn2 siRNA. (**B**) Quantification of transferrin uptake in NRK cells treated with control or Dyn2 siRNA. Data represent mean ± SEM of fluorescence intensity/microscopic field. ***p<0.001. Inset images: red, transferrin; blue, DAPI-stained nuclei. (**C**) Quantification of FM1–43 influx in NRK cells treated with control or Dyn2 siRNA, and exposed or not to SLO ± Ca$^{2+}$. Data represent mean ± SEM of fluorescence intensity/cell. The results are representative of three independent experiments. (**D**) TEM of Dyn2 siRNA-treated cells exposed to SLO or (SM) for 60 s, fixed and stained with ruthenium red. Small arrows: unstained caveolae. Wide arrows: stained clathrin-coated vesicles and elongated endocytic structures connected to the PM. Bars: 100 nm. (**E**) Western blot with anti-dynamin-1, anti-dynamin-2, and anti-actin (loading control) in lysates of cells derived from dynamin-1 and -2 conditional double knockout mice, induced with tamoxifen (DKO) or not induced (Control). (**F**) Quantification of transferrin uptake in

*Figure 7. Continued on next page*

*Figure 7. Continued*

control or dynamin-1-2 DKO cells. Data represent mean ± SEM of fluorescence intensity/microscopic field. **p=0.003. Inset images: red, transferrin; blue, DAPI-stained nuclei. (**G**) Time-lapse imaging of FM1–43 and PI influx during permeabilization with SLO ± $Ca^{2+}$ in control and dynamin-1 and -2 DKO cells, (**H**) Quantification of CTxB-A488 uptake by control and dynamin-1-2 DKO cells exposed or not to SLO or SM for 180 s. Data represent mean ± SEM of fluorescence intensity/microscopic field. *p=0.014, ***p<0.001 (comparison with respective control conditions), unpaired Student's *t* test. All results in this figure are representative of three or more independent experiments.

myotubes inhibits caveolae formation and PM repair. Collectively, our study identifies caveolar vesicles as dynamic endocytic structures that play a key role in the restoration of PM integrity.

Our findings have important implications for the understanding of muscular dystrophy. Mutations in the muscle-specific caveolin Cav3 cause at least five forms of muscle pathology, including limb girdle muscular dystrophy 1C, rippling muscle disease, distal myopathy, hyperCKemia, and hypertrophic cardiomyopathy (*Gazzerro et al., 2010*). Mutations in PTRF/cavin1 also lead to muscular dystrophy and cardiac dysfunction (*Rajab et al., 2010*), and overexpression of this caveolae-associated protein rescues membrane repair defects in dystrophic muscle (*Zhu et al., 2011*). In earlier studies these effects were not attributed to caveolae assembly and internalization, but rather to an independent role of caveolin and cavin molecules in regulating the traffic of unrelated proteins involved in muscle fiber repair, such as dysferlin and MG53 (*Hernández-Deviez et al., 2008*; *Hayashi et al., 2009*; *Cai et al., 2009b*; *Zhu et al., 2011*). Our study sheds important new light on this issue, by showing that caveolar vesicles are directly involved in the endocytic mechanism by which cells remove wounds from their PM.

Duchenne muscular dystrophy fibers were shown in numerous studies to contain elevated numbers of caveolae, but this important finding was also attributed to secondary effects of alterations in Cav3 expression levels (*Bonilla et al., 1981*; *Repetto et al., 1999*). However, it is important to note that Duchenne muscular dystrophy is caused by mutations in dystrophin, a main component of the dystroglycan complex that confers stability to the sarcolemma (*Cohn and Campbell, 2000*). Duchenne dystrophic fibers are very susceptible to contraction-induced wounding, and are likely to undergo repeated cycles of injury and repair—a process that we now show to involve intracellular accumulation of caveolar vesicles, in several cell types including C2C12 myotubes and primary muscle fibers. Thus, our findings provide a novel explanation for the accumulation of caveolae within fragile muscle fibers that is fully consistent with a direct role of caveolar endocytosis in sarcolemma resealing. Our results are also in agreement with the elevated number of caveolar profiles observed in cell types subject to chronic membrane stress, such as endothelial cells and adipocytes (*Parton and Simons, 2007*).

Caveolae are thought to be dynamic structures, but the signals leading to their assembly and internalization remain a matter of debate (*Lajoie and Nabi, 2010*; *Sandvig et al., 2011*). Caveolae are enriched in cholesterol and glycosphingolipids, and contain the abundant surface lipid sphingomyelin that generates ceramide after cleavage of its phosphorylcholine head group by SM (*Liu and Anderson, 1995*; *Parton and Simons, 2007*). Exogenously added glycosphingolipids selectively stimulate caveolar endocytosis (*Sharma et al., 2004*), and the ceramide core of glycosphingolipids was identified as an important determinant of caveolae internalization (*Singh et al., 2003*). However, the mechanism by which glycosphingolipids modulate caveolar endocytosis remained unclear, because a non-hydrolyzable synthetic glycosphingolipid analog was reported to have the same effect (*Sharma et al., 2004*). Our study clarifies this issue, by directly demonstrating that treatment of the outer leaflet of the PM with the ceramide-generating enzymes ASM or SM is sufficient to induce internalization of vesicles with properties of caveolae. In agreement with our findings, ASM (*Opreanu et al., 2011*) and ceramide (*Liu and Anderson, 1995*; *Bilderback et al., 1997*; *Czarny et al., 2003*) were detected in caveolae-enriched membrane fractions.

Collectively, our results support a novel model proposing that injury to the PM triggers $Ca^{2+}$ influx, exocytosis of lysosomes, ASM release and generation of surface-associated ceramide—an event that facilitates caveolae internalization and lesion removal/repair (*Figure 13A,B*). We consistently observed a strong inhibition of PM repair in cells acutely depleted of Cav1 by RNAi (this study) or depleted in cholesterol (*Idone et al., 2008*). These findings are in full agreement with our additional lines of evidence implicating caveolar vesicles as endocytic carriers responsible for lesion removal from the PM. However, it is also important to note that a complex cross-talk exists between caveolar and other

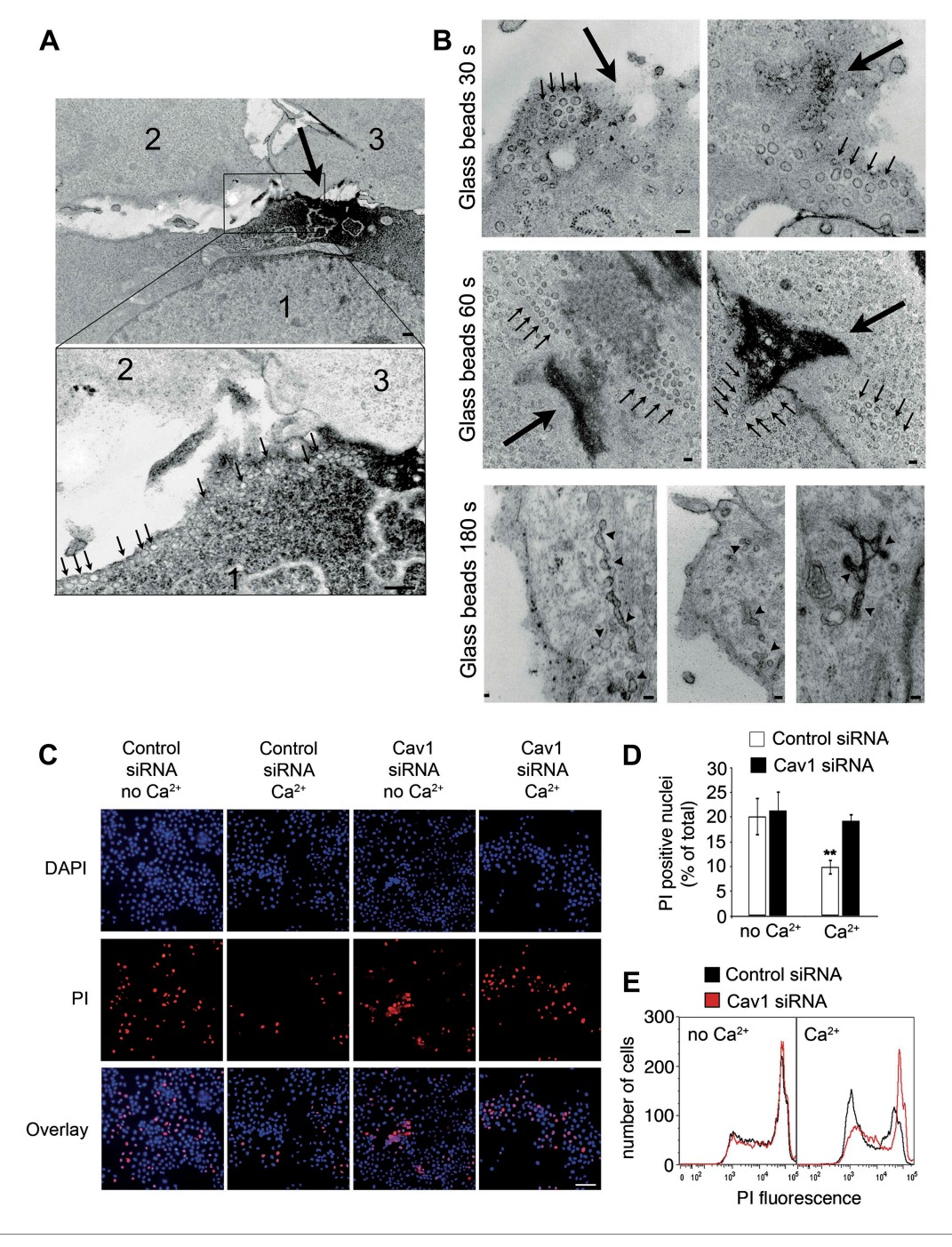

**Figure 8**. Caveolae accumulate at sites of mechanical wounding, and Cav1 is required for mechanical wound repair. (**A**) TEM of NRK cells wounded with glass beads and stained with ruthenium red during fixation. Numerous caveolae-like vesicles (arrows, lower magnified image) are visible near the wound, identified by ruthenium red influx in cell #1 (large arrow, upper image). Two non-wounded cells are present in the same field (#2 and #3, upper image). Bar: 500 nm. (**B**) TEM of NRK cells wounded with glass beads and stained with ruthenium red. Small arrows: clusters of caveolae-like vesicles. Large arrows: wound sites. Arrowheads: merged caveolae-like vesicles connected to the PM. Bars: 100 nm. (**C**) NRK cells treated with control or Cav1 siRNA, wounded with glass beads ± $Ca^{2+}$ and stained with PI (red) and (blue). Bar: 50 μm. (**D**) Quantification of PI positive nuclei in **D**. Data represent the mean ± SEM of the %PI positive cells/field. **p=0.0012 (compared to no $Ca^{2+}$ conditions), unpaired Student's *t* test. The results are representative of two independent experiments. (**E**) FACS quantification

*Figure 8. Continued on next page*

*Figure 8. Continued*

of PI staining in NRK cells treated with control or Cav1 siRNA, mechanically wounded by scraping from the dish ± Ca$^{2+}$. The results are representative of two independent experiments.

The following figure supplements are available for figure 8:

**Figure supplement 1**. Clusters of caveolae are observed next to sites of mechanical wounding.

cholesterol-dependent, clathrin-independent endocytic pathways. Thus, non-caveolar cholesterol/sphingolipid raft-dependent endocytic carriers may also play a role in removing lesions from the PM under some conditions, such as in Cav1-deficient cells, which were shown to upregulate non-caveolar endocytic pathways (*Le et al., 2002*; *Nichols, 2003*; *Singh et al., 2003*).

The GTPase dynamin (*Guha et al., 2003*; *Hansen and Nichols, 2009*) was detected on caveolae necks (*Henley et al., 1998*; *Oh et al., 1998*), and proposed to be required for some forms of caveolar endocytosis (*Henley et al., 1998*; *Oh et al., 1998*; *Yao et al., 2005*). However, more recent studies using dominant-negative forms of dynamin-2 (*van Deurs et al., 2006*; *Liu et al., 2008*) or cells deficient in both dynamin-1 and dynamin-2 (*Ferguson et al., 2009*) failed to observe inhibition of caveolar endocytosis and the accumulation of PM-connected caveolar intermediates expected from a fission block. These findings led to suggestions that indirect effects on the actin cytoskeleton or sequestration of essential factors distinct from dynamin might explain the inhibition in caveolar endocytosis by dominant-negative dynamin (*Liu et al., 2008*). After extensive analysis of cells depleted in dynamin-1 and dynamin-2 we did not observe any defects in caveolar endocytosis in response to PM wounding, suggesting that this process may be dynamin-independent. Not all forms of endocytosis require the GTPase dynamin (*Guha et al., 2003*; *Hansen and Nichols, 2009*), and the existence of different forms of caveolar endocytosis has been proposed (*Le and Nabi, 2003*; *Singh et al., 2003*). Given that ceramide accumulation is sufficient to induce membrane invagination and inward budding of membranes (*Holopainen et al., 2000*; *Trajkovic et al., 2008*), it is an intriguing possibility that sphingomyelin hydrolysis and ceramide generation may bypass the requirement for dynamin in the fission of caveolar vesicles associated with PM repair.

When imaged several minutes after PM injury, endocytic vesicles triggered by Ca$^{2+}$ influx appear as large, uncoated vesicles containing the early endosome marker EEA1 (*Keefe et al., 2005*; *Idone et al., 2008*). A recent study showed that internalized SLO moves to the perinuclear area in endocytic vesicles that gradually increase in size and merge with late endosomes/lysosomes, where ubiquitinated toxin is degraded in an ESCRT-dependent manner (*Corrotte et al., 2012*). Strikingly, in this study we found that cell injury or exposure to purified SM initially generates homogeneous <80 nm vesicles with a morphology typical of caveolae, which then rapidly merge originating larger, irregular intracellular compartments. Interestingly, wounded cells accumulate larger and more abundant endocytic vesicles after the cortical cytoskeleton is disrupted with cytochalasin D (*Tam et al., 2010*), a condition that enhances caveolae motility (*Thomsen et al., 2002*). Thus, removal of obstacles presented by the cortical cytoskeleton may facilitate internalization and intracellular merging of caveolae. Supporting this view, we observed reduced levels of EHD2, the ATPase proposed to link caveolae to actin filaments (*Moren et al., 2012*; *Stoeber et al., 2012*), on caveolar vesicles that move deeper into cells after injury or exposure to SM.

In several instances caveolae appeared to merge intracellularly while still connected to the PM, particularly when cells were mechanically injured. We envision that the rapid intracellular merging of surface-connected caveolae, possibly enhanced by the higher levels of Ca$^{2+}$ flowing through large lesions, might generate forces on the PM that constrict and ultimately reseal wounds (*Figure 13B*). Such 'bunch of grapes' pattern of caveolar vesicles is also observed in cells not subjected to a specific wounding procedure (*Fra et al., 1995*). However, it is important to note that lifting cells from dishes and other routine cell handling steps often results in PM wounding and Ca$^{2+}$ influx. Detaching cells from their substratum was actually reported to trigger rapid and robust caveolae internalization (*Muriel et al., 2011*). Our observations suggest a model where caveolae-derived branched endocytic structures still tethered to the PM would be generated as a consequence of the vigorous Ca$^{2+}$ influx and localized release of lysosomal ASM triggered by large wounds. These branched caveolae-derived structures appear to accumulate around the periphery of the PM wounds, and as their deeper portions

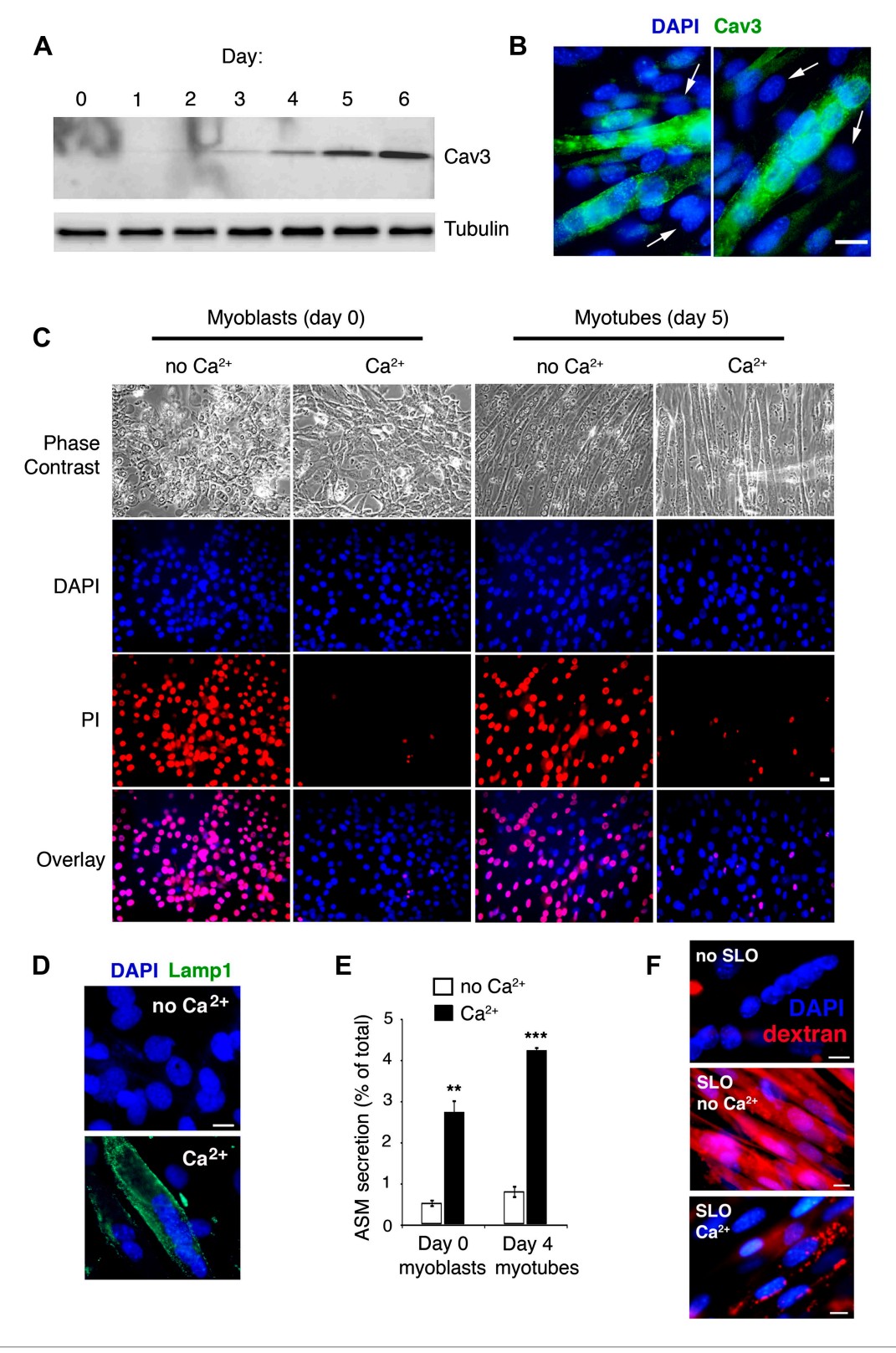

**Figure 9**. Cav3 expression, Ca²⁺-dependent sarcolemma repair, lysosomal exocytosis and endocytosis in C2C12 myoblasts/myotubes. (**A**) Western blot with anti-Cav3 or anti-tubulin (loading control) antibodies showing that differentiation of C2C12 myoblasts induced by serum starvation leads to a gradual enrichment of the cultures in

*Figure 9. Continued on next page*

*Figure 9. Continued*

myotubes expressing the caveolin isoform Cav3. (**B**) Immunofluorescence of C2C12 cultures at day 5 after serum starvation with anti-Cav3 antibodies. Green: Cav3 staining in myotubes. Blue: DAPI-stained nuclei. Arrows point to DAPI-stained nuclei of Cav3-negative myoblasts. Bar: 100 nm. (**C**) C2C12 cells at day 0 (myoblasts) and day 5 (enriched in myotubes) after serum starvation, permeabilized with 400 ng/ml SLO and stained with PI (membrane impermeable) and DAPI (membrane permeable) after 4 min with or without $Ca^{2+}$. Bar: 10 µm. The results in **A**–**C** are representative of three independent experiments. (**D**) Immunofluorescence with anti-Lamp1 in live C2C12 myotubes 15 min after permeabilization with SLO ± $Ca^{2+}$. Green: Lamp1 luminal epitope. Blue: DAPI-stained nuclei. Bar: 10 µm. The results are representative of two independent experiments. (**E**) ASM activity secreted by undifferentiated myoblasts (day 0) or myotube-enriched C2C12 cultures (day 4) after SLO±$Ca^{2+}$ for 240 s. Data represent mean ± SEM of triplicates. **p=0.002, ***p<0.001, unpaired Student's *t* test . (**F**) Myotubes permeabilized or not with SLO±$Ca^{2+}$ for 240 s in the presence of Texas Red-dextran. Red: dextran. Blue: DAPI-stained nuclei. Bars: 10 µm. The results are representative of three independent experiments.

merge intracellularly, constriction forces may be generated on the PM facilitating bilayer resealing. This model predicts that when the complex caveolar structures pinch off from the PM, large intracellular vesicles would be generated next to injury sites (*Figure 13B*). This scenario is fully consistent with previous EM observations of large vesicles close to wound sites (*McNeil and Steinhardt, 2003*). These large intracellular vesicles were at the time interpreted as exocytic 'patches', responsible for resealing the wound. Here we propose a very different scenario for the repair of mechanical injury to the PM: After $Ca^{2+}$-triggered lysosomal exocytosis and ASM release, caveolae-derived vesicles would move into cells while undergoing a homotypic fusion process, which would ultimately result in large compartments of endocytic origin (as opposed to a large exocytic patch, as proposed in earlier models—[*McNeil and Steinhardt, 2003*]).

A vesicle internalization/constriction mechanism for PM resealing predicts that significant local tension would be generated on PM sites close to the wound. This tension may be dissipated by the localized exocytosis of lysosomes, and/or by the recently demonstrated process of flattening of individual caveolae in response to membrane stress (*Sinha et al., 2011*). An intriguing possibility is that mechanical wounds may actually occur preferentially in areas where caveolae are present, because membranes may be rendered more fragile by the flattening of caveolae that follows a mechanical stretch (*Sinha et al., 2011*). Caveolae are extremely abundant in cells that are under mechanical stress in vivo, such as smooth muscle fibers and endothelial cells (*Parton and Simons, 2007*). Our findings now suggest that frequent PM injury and repair may represent the mechanism responsible for upregulation of the caveolar population in these cells.

In summary, our study provides a novel framework for understanding how endocytosis functions in the maintenance and restoration of PM integrity, particularly in tissues under high mechanical stress such as skeletal muscle. Importantly, this work also identifies caveolar endocytosis as a pathway that may be directly affected in several types of muscular dystrophy. Resealing of injured muscle fibers requires muscle-specific proteins such as dysferlin and MG53, which accumulate in vesicles close to sites of injury and have been assumed to facilitate formation of an exocytic membrane patch (*Cai et al., 2009a*). In light of our findings and of the newly uncovered role of endocytosis in PM repair, it will be of great interest to determine whether these muscle-specific proteins participate in wound resealing by facilitating caveolar assembly/internalization, and lesion removal by endocytosis.

## Materials and methods

### Cell culture and treatments

NRK and HeLa cells were cultured at 37°C in 5% $CO_2$ in high glucose DME containing 10% heat-inactivated FBS and penicillin/streptomycin (Invitrogen, Grand Island, NY). The C2C12 mouse myogenic cell line (CRL-1772), a subclone of mouse skeletal muscle C2 cells (*Blau et al., 1983*), was obtained from the American Type Culture Collection (Rockville, MD). C2C12 were grown to confluency at 37°C and 5% $CO_2$ in DME supplemented with 10% FBS, 100 units/ml penicillin/streptomycin. The medium was then changed to DME containing 2% horse serum and 100 U/ml penicillin/streptomycin to trigger myogenic differentiation. The cells were maintained in this medium for 4–7 days, with fresh medium added every second day.

The tamoxifen-inducible DKO fibroblast line generated from dynamin-1 and 2 double conditional knockout mice (*Ferguson et al., 2009*) was provided by De Camilli P, Yale University and cultured at 37°C

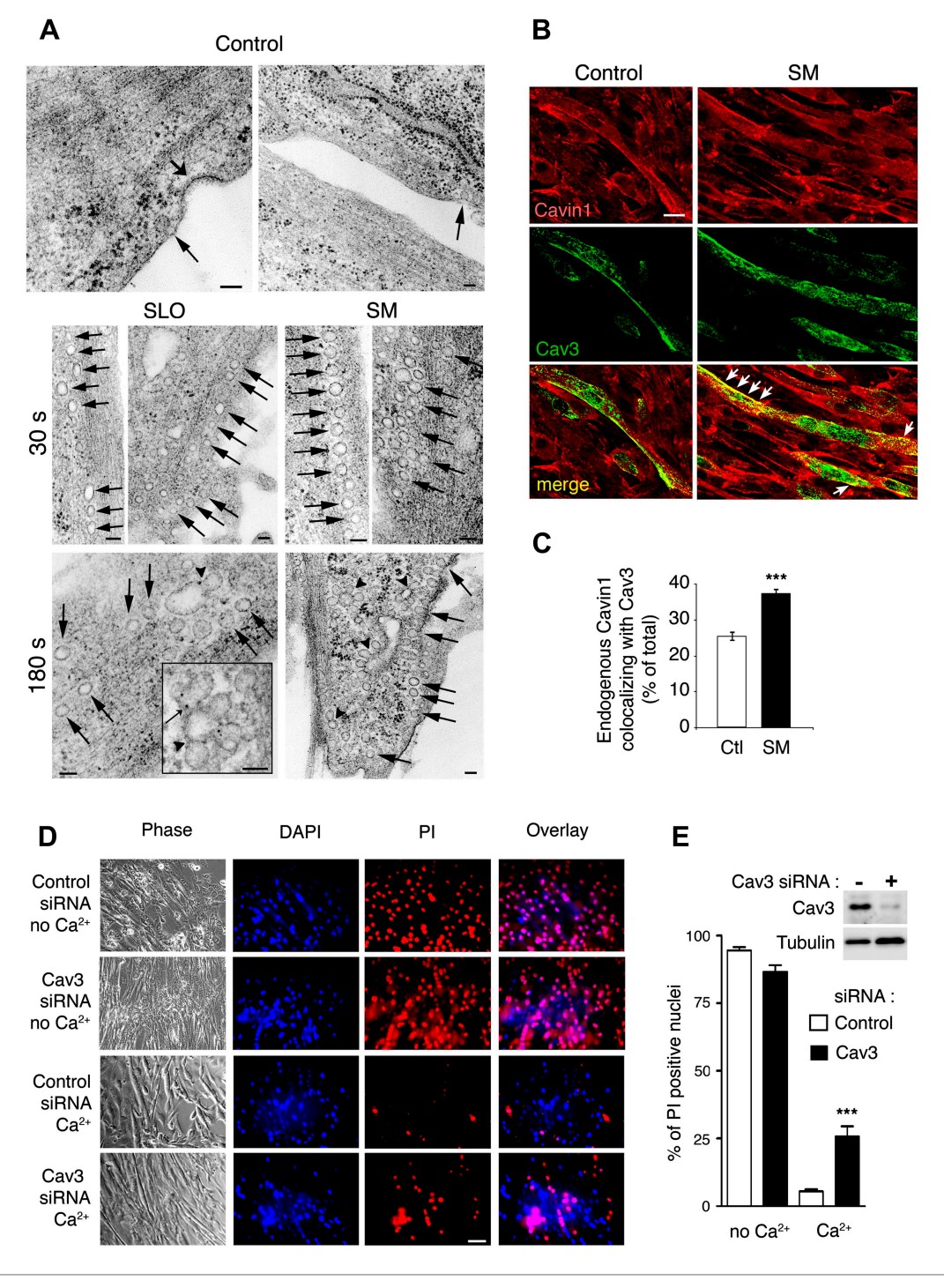

**Figure 10**. Caveolae-like vesicles accumulate in C2C12 myotubes exposed to sphingomyelinase and SLO, and resealing after injury depends on Cav3. (**A**) TEM of myotubes untreated (control) or exposed to SLO or SM. Arrows: caveolae-like vesicles. Arrowheads: merged caveolae-like vesicles. Wide arrow: clathrin-coated pit. Inset: higher magnification showing luminal BSA-gold (thin arrow). Bars: 100 nm. The results are representative of four independent experiments. (**B**) Confocal images of immunofluorescence with anti-cavin1 (red) and anti-Cav3 (green) in C2C12 myotubes treated or not with SM for 30 s. Arrows show colocalization of cavin1 and Cav3 staining. Bar: 10 μm. The results are representative of two independent experiments. (**C**) Quantification of the fraction of all cavin1 colocalizing with Cav3 in **E**. Data represent mean ± SEM of the colocalization coefficient/myotube. ***p<0.001, unpaired Student's *t* test. (**D**) Images of

*Figure 10. Continued*

myotube-enriched cultures treated with control or Cav3 siRNA, exposed to SLO±Ca$^{2+}$ for 240 s and stained with PI and DAPI. PI (red) and DAPI (blue). Bar: 50 μm. The results are representative of four independent experiments. (**E**) Western blot of Cav3 or tubulin (loading control), and quantification of PI positive nuclei in an assay performed as in (**D**). Data represent the mean of three independent experiments ± SEM. ***p<001, unpaired Student's *t* test.

in 5% CO$_2$ in high glucose DME containing 10% heat-inactivated FBS and penicillin/streptomycin (Invitrogen). To induce dynamin-1 and 2 depletion, cells were incubated for 2 days in medium containing 3 μM tamoxifen (Sigma, St. Louis, MO) followed by 3–4 days of 300 nM tamoxifen before seeding for experiments.

*Flexor digitorum brevis* muscle fibers (***Cai et al., 2009a***) were surgically isolated from euthanized male C57Bl/6 mice in a Tyrode solution containing 140 mM NaCl, 5 mM KCl, 2.5 mM CaCl$_2$, 2 mM MgCl$_2$ and 10 mM HEPES (pH 7.2), and incubated in the same solution containing 2 mg/ml type I collagenase (Sigma) in an orbital shaker at 100 rpm for 60 min, followed by gravity sedimentation for 5 min at 37°C. After discarding the supernatant, Tyrode solution was added and the pellet gently resuspended, followed by a second round of gravity sedimentation for 1 min to remove large tissue aggregates. The supernatant containing isolated fibers was transferred to another tube, allowed to sediment for 5 min, resuspended in DME 10% FBS and subjected to the various treatments before PI influx assays and/or fixation for TEM.

## Antibodies

Immunoblot, immunofluorescence and immuno-EM assays were performed using rabbit anti-GFP to detect GFP-SLO (Invitrogen), rabbit anti-Cavin1/PTRF (Abcam, Cambridge, MA), mouse anti-Cav-1 and -3 (BD transduction Laboratories, San Jose, CA), rabbit anti-Cav1 (Santa Cruz, Dallas, TX), rat anti-mouse Lamp1 (1D4B mAb, Developmental Studies Hybridoma Bank, Iowa City, IA), mouse anti-ceramide (mAb 15B4; Sigma), mouse anti-tubulin (Sigma), mouse anti-actin (Sigma), rabbit anti-dynamin-2 (Abcam), rabbit anti-dynamin-1 (Epitomics, Burlingame, CA), and goat anti-EHD2 (Abcam).

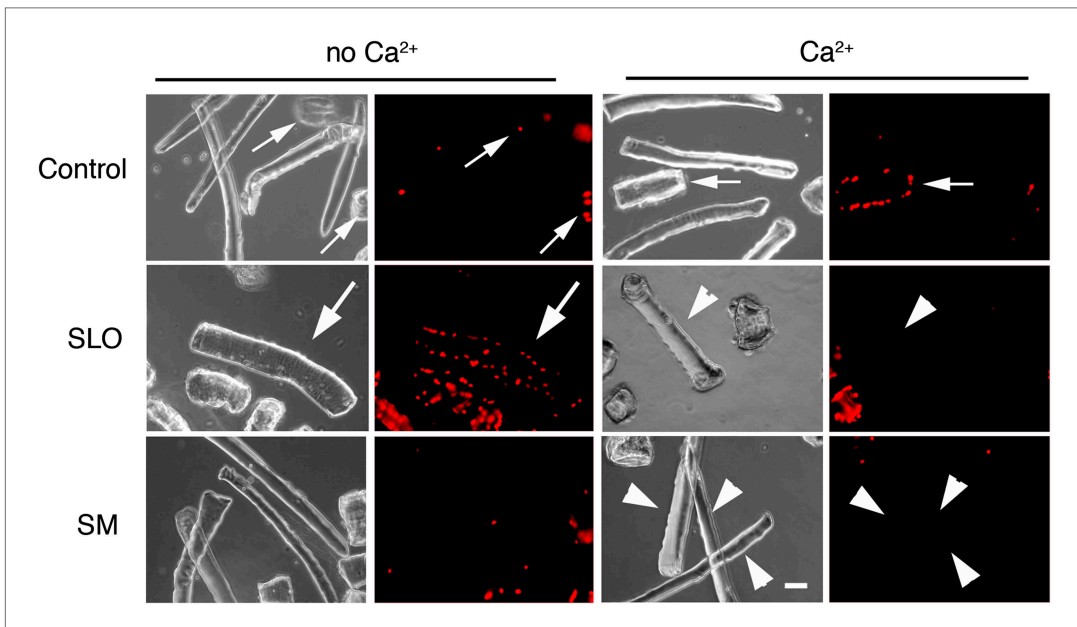

**Figure 11**. Primary muscle fibers are sensitive to SLO permeabilization and reseal in the presence of Ca$^{2+}$. *Flexor digitorum brevis* mouse muscle fibers treated or not with 400 ng/ml SLO or 50 mU/ml SM in the presence or absence of Ca$^{2+}$ and stained with PI (red) after 30 s. Small arrows point to PI positive fibers that were injured during dissection and failed to reseal. Arrowheads point to PI-negative fibers that resealed after SLO+Ca$^{2+}$, or were not injured by the SM treatment. Large arrows point to a PI-positive fiber that was injured by SLO and failed to repair without Ca$^{2+}$. Bar: 50 μm.

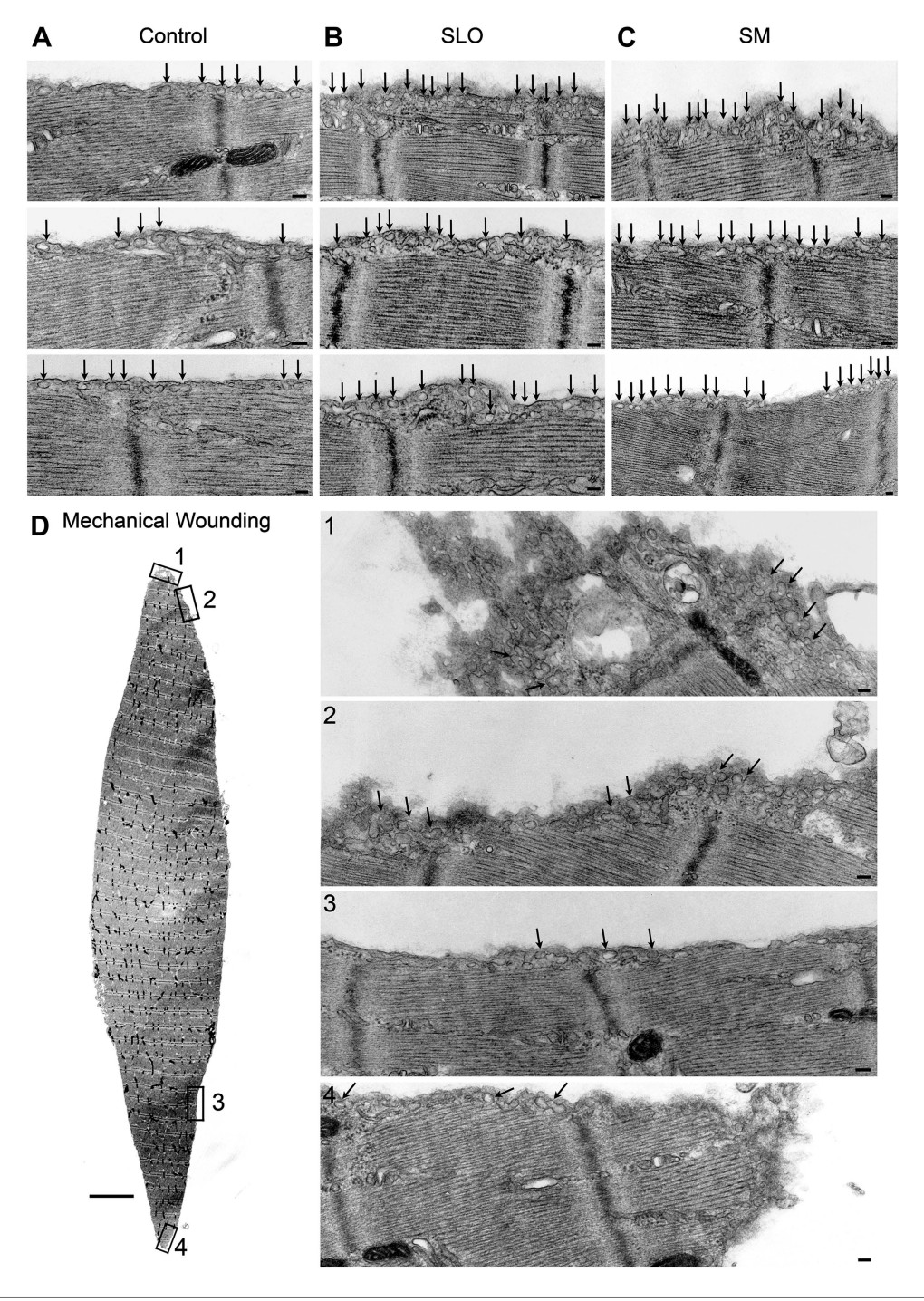

**Figure 12**. Caveolae accumulate in primary mouse muscle fibers after exposure to sphingomyelinase or sarcolemma injury. (**A**–**C**) TEM of *flexor digitorum brevis* fibers untreated (**A**) or exposed for 300 s to SLO (**B**) or SM (**C**). Three examples are shown for each. Arrows: single or merged caveolae-like vesicles. Bars: 100 nm. The results are representative of four independent experiments. (**D**) TEM of fiber fixed immediately after dissection, showing mechanical damage (site #1). Bar: 5 μm. Panels 1–4 show enlarged images of the regions indicated in the whole fiber image. Arrows: single or merged caveolae. Bars: 100 nm. The results are representative of two independent experiments.

The following figure supplements are available for figure 12:

**Figure supplement 1**. Accumulation of caveolae-like vesicles in mechanically injured *flexor digitorum brevis* muscle fibers.

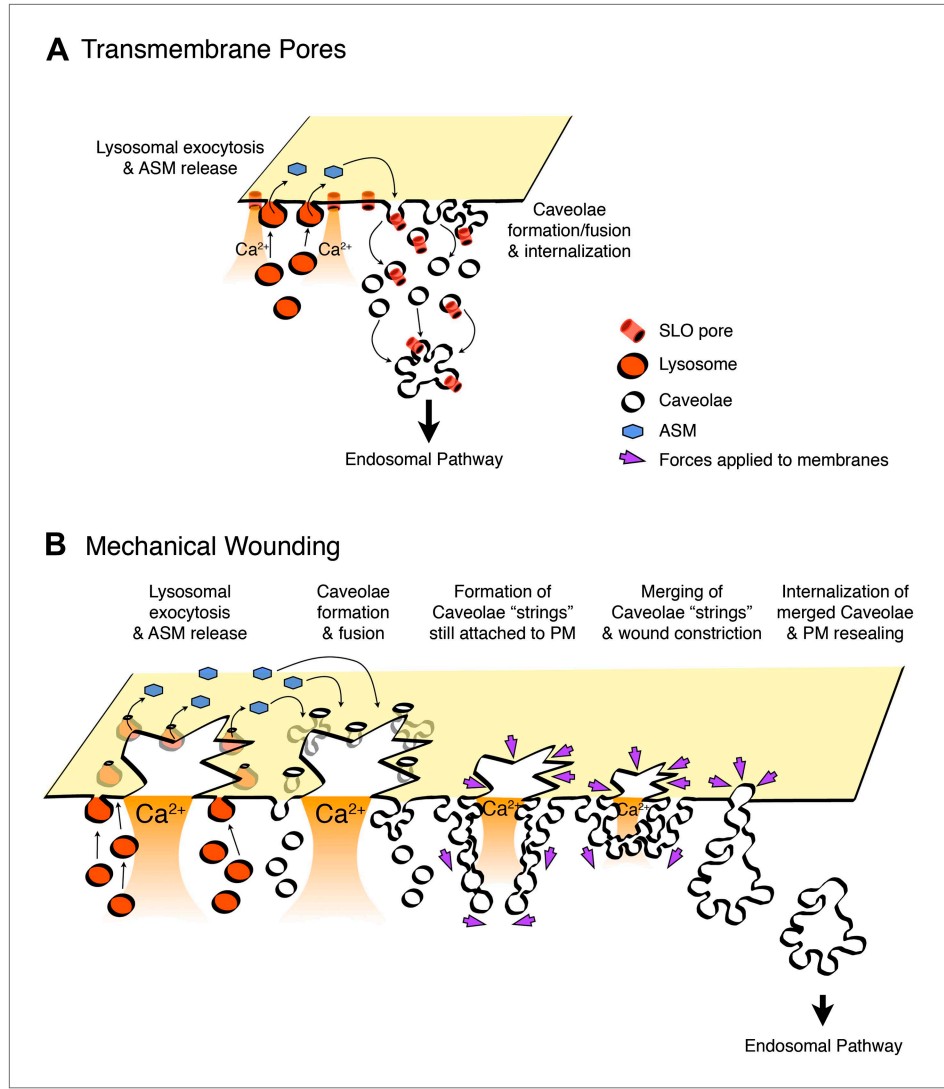

**Figure 13**. Model for PM repair mediated by caveolar endocytosis. Permeabilization with transmembrane toxin pores (**A**) or mechanical wounding (**B**) triggers $Ca^{2+}$ influx, exocytosis of lysosomes, release of ASM, and generation of ceramide at the PM outer leaflet, a process that promotes caveolae internalization and fusion. Toxin pores would be removed from the PM by caveolar endocytosis (**A**), while larger breaches on the lipid bilayer would be gradually constricted and resealed as a results of forces generated on the PM by the intracellular clustering, fusion and internalization of merged/branched caveolar structures (**B**).

## Transcriptional silencing and expression of mRFP-Cav1

NRK or HeLa cells (50% confluency) and C2C12 myoblasts (70% confluency, 48–72 hr after FBS removal) in reduced serum DME without penicillin/streptomycin were transfected with Lipofectamine RNAiMax (Invitrogen) and 960 pmol of medium-content control, Cav1, Cav3, ASM or dynamin-2 (Dyn2) stealth siRNA duplexes, according to the manufacturer's instructions (Invitrogen). After 48 hr–72 hr cells were treated with SLO, SM or glass beads and processed for various assays. HeLa cells (50% confluency) in MatTek glass-bottom dishes containing reduced serum DME without penicillin/streptomycin were transfected with Lipofectamine 2000 (Invitrogen) and 1 μg of mRFP-Cav1 plasmid per dish, according to manufacturer's instructions (Invitrogen). After 24 hr cells were processed for live imaging. The correct targeting of mRFP-Cav1 to caveolae was confirmed by acquiring Z stack images (0.13 μm Z steps) in a confocal Leica SPX5 microscope with a 63 × 1.4 N.A. oil objective of transfected HeLa cells fixed, permeabilized and stained with anti-Cav1 antibodies.

## Transmission electron microscopy (TEM) and cryo-immuno EM

Cells were pre-treated with 200–400 ng/ml SLO (NRK and HeLa) or 800 ng/ml (C2C12 myotubes and *flexor digitorum brevis* mouse muscle fibers) for 5 min at 4°C and further incubated for various time points at 37°C in DME containing BSA-gold (OD 520 nm = 200; [*Ferguson et al., 2009*]), or incubated in DME containing 10 µg/ml human recombinant acid sphingomyelinase (ASM) (*He et al., 1999*) or 50 mU/ml of *B. cereus* sphingomyelinase (SM) (Invitrogen), before being processed for TEM as previously described (*Rodríguez et al., 1997*). EM images were acquired randomly or along the PM. Quantifications were performed by counting all vesicles containing BSA-gold in several cell sections/sample, or by counting all caveolae-like vesicles with a diameter of less than 80 nm (measured with the line tool of ImageJ, NIH) in 20–30 cell sections/sample or in 40 images/sample. The relative amount of < and >80 nm vesicles was determined in cells treated with 200 ng/ml SLO for 30, 60 and 180 s in the presence of BSA-gold. Vesicle area was measured using the outline function of ImageJ and the number of gold particles/vesicle was determined in 14–47 cell sections. To visualize mechanical wounding with glass beads in NRK cells or to assess complete separation of vesicles from the PM in control or Dyn2 siRNA-treated NRK cells treated with SLO or SM, cells were fixed in 2% glutaraldehyde in 0.1M cacodylate and 0.05% ruthenium red for 1 hr at room temperature before washing and processing for TEM as described in *Parton et al. (2002)*. EM images were blinded before quantification and scored independently by two investigators. For cryo-immuno-EM, cells were treated as described above for TEM and fixed in 4% PFA, 0.25 M HEPES and 0.1% glutaraldehyde for 1 hr at room temperature and processed for immuno-gold labeling of Cav1, GFP-SLO or ceramide as described in *Czibener et al. (2006)*. Ceramide staining was quantified by drawing a line along the PM using the ImageJ brush tool set to 200 nm diameter, and counting all gold particles inside the brush tool area in all membrane sections (cell section areas ranged from 6 to 16 µm$^2$ and the data were normalized to particles/µm$^2$). To assess localization of Cav1 and GFP-SLO over time during PM repair, Cav1 and GFP-SLO positive structures were quantified by counting all <80 nm vesicles or >80 nm vesicles positive for anti-Cav1 alone, anti-GFP alone or both, as well as flat PM areas positive for anti-GFP. All antibody-stained structures were quantified in 80 random microscopic fields for each sample. The relative Cav1 localization between flat PM stretches and vesicular profiles was quantified by counting 25–72 flat PM segments and 256–346 vesicular profiles positive for anti-Cav1 in 80 images/sample. For all antibodies, titrations were performed and specificity was assessed using isotype control antibodies before imaging and quantifications, according to standard procedures from the Yale University Center for Cell and Molecular Imaging.

## Alexa 488-SLO endocytosis assays

To analyse SLO endocytosis during PM repair by flow cytometry, a single cysteine mutant of SLO (SLO$^{G66C}$, generated by replacing Gly66 by a cysteine in a cysteine-less derivative of SLO, kindly provided by Dr. R Tweten, U Oklahoma) was labeled at the N-terminus with a thiol-reactive AlexaFluor-488 C5 maleimide (Life Technologies, Frederick, MD) according to the manufacturer's instructions. Briefly, 500 µl of a 50 µM SLO solution (in PBS no Ca$^{2+}$) were incubated with 10 mM of reactive dye for 2 hr at room temperature. After reaction, the labeled SLO was separated from unbound dye by gel filtration. Subconfluent NRK cells were treated or not with control or Cav1 siRNA for 48 hr, trypsinized, counted and diluted to $1,5 \times 10^5$ cells/250 µl for flow cytometry. Cells were incubated at 4°C for 5 min with increasing concentrations (0.9–2.1 µg/ml) of Alexa 488-SLO in PBS supplemented with 5.5 mM D-glucose with or without Ca$^{2+}$ for PM binding, and then transferred or not to 37°C for 5 min to induce PM repair, followed by transfer to 4°C to stop the process. Cells were analyzed by flow cytometry (FACSCanto, Becton Dickinson, Sparks Glencoe, MD) and A488 fluorescence was assessed before and after adding 10 µg/ml of rabbit anti-Alexa Fluor 488 quenching antibody (Life Technologies) for 2 min. PI (50 µg/ml) was added to all samples at the end of the assay to assess levels of PM repair. Data were analyzed using Flo-Jo software (Three Star, Inc, Ashland, VA).

## Live time-lapse imaging of FM1–43 influx and GFP-SLO internalization

Subconfluent NRK cells treated with control, Cav1 or Dyn2 siRNA, or DKO MEFs induced or not with tamoxifen were plated on glass-bottom dishes (MatTek, Ashland, MA), pre-incubated with 200 ng/ml of SLO for 5 min at 4°C, transferred to a LiveCell System chamber (Pathology Devices, Westminster, MD) at 37°C with 5% CO$_2$, and exposed to pre-warmed DME containing or not Ca$^{2+}$ and 4 µM (NRK cells) or 8 µM (dynamin-1-2 DKO cell line) FM1–43 (Invitrogen) and SLO, as previously described (*Idone et al., 2008*).

Spinning disk confocal images were acquired for 4 min at 1 frame/3 s using an UltraVIEW VoX system (PerkinElmer, Waltham, MA) attached to an inverted microscope (Eclipse Ti; Nikon Instruments, Melville, NY) with a 40 × NA 1.3 objective (Nikon) and a CCD camera (C9100–50; Hamamatsu Photonics, Bridgewater, NJ). Quantitative analysis of fluorescence in a defined intracellular area was performed using Volocity Suite (PerkinElmer). For live imaging of GFP-SLO internalization, HeLa cells were cultured on glass-bottom dishes (MatTek) and transfected with mRFP tagged Cav1 (mRFP-Cav1) for 24 hr using lipofectamine 2000 (Invitrogen). Cells expressing mRFP-Cav1 were pre-incubated for 5 min on ice with 800 ng/ml of GFP-SLO (generated as described in *Idone et al. [2008]*), and 4°C DME with $Ca^{2+}$ was added to induce PM repair after warming to 37°C on a heated stage, followed by imaging for 5 min at 1 frame/2 s as previously described for FM1–43 imaging. Since the amount of SLO fluorescence that becomes associated with cells under conditions that allow PM repair is limited, to bring the signal out from the noise floor inherent to the imaging system all datasets were processed using NIS-Elements software (Nikon Instruments). First, drift correction was performed on datasets to account for drift with temperature changes at high magnification. In order to improve the signal to noise ratio and minimize the effects of the spurious noise, a 3-frame rolling average was employed followed by a regional maximum detection, which compares a pixel (or group of pixels) to its neighboring region and determines where there is a significant difference. Routine scaling of the image resulted in the contrast viewed in the included datasets (*Videos 1, 2 and 3*). Volocity Suite (Perkin Elmer) was used to draw lines beneath the PM to record fluorescence intensity levels of GFP-SLO and mRFP-Cav1 and create kymographs displaying fluorescence levels over time (Y axis) for each pixel along the line (X axis). Colocalization of GFP-SLO and mRFP-Cav1 was analyzed and displayed as positive PDM (product of the differences from the mean).

## Immunofluorescence and TIRF microscopy

HeLa cells were treated or not with 450 ng/ml SLO or 50 mU/ml SM for 30 or 60 s, fixed, permeabilized with 0.05% saponin in PBS containing 1% BSA and immunostained overnight for Cav1 (Santa Cruz) and EHD2 (Abcam) diluted 1:500 in permeabilization buffer, followed by 1 hr incubation with secondary antibodies conjugated with Alexa fluor 488 (anti-rabbit) or Alexa fluor 546 (anti-goat). Images were acquired using a Nikon laser TIRFm system on an inverted microscope (Nikon TE2000-PFS) equipped with a 63 × NA 1.49 Apochromat TIRF objective (Nikon Instruments), a Coolsnap HQ2 charge-coupled device camera (Roper Scientific, Sarasota, FL), and two solid-state lasers of wavelengths 491 and 561 nm. AF488 and AF546 images were acquired sequentially and analysis of fluorescence intensity for Cav1 staining was performed on 53–79 cells per sample using Andor iQ software (Andor Technology, Belfast, UK). Colocalization of Cav1 (AF488) with EHD2 (AF546) was quantified on 65–82 cells/sample using Volocity Suite (PerkinElmer) after applying thresholding to each channel.

## Western blot

Following extraction proteins were separated on 8, 10 or 12% SDS-PAGE gels and blotted on nitrocellulose membranes using the Trans-Blot Transfer system (Bio-Rad Laboratories, Hercules, CA) overnight at 30 V, or for 2 hr at 95 V. After incubation with the primary antibodies and peroxidase conjugated secondary antibodies, detection was performed using Supersignal West Pico Chemiluminescent Substrate (Thermo Scientific, Waltham, MA) and a Fuji LAS-3000 Imaging System with Image Reader LAS-3000 software (Fuji, Edison, NJ).

## CTxB-A488 uptake assay and quantification

Dynamin-1-2 DKO MEFs were incubated or not with 400 ng/ml of SLO for 5 min at 4°C followed by incubation for 3 to 10 min at 37°C in DME with $Ca^{2+}$ and 5 μg/ml of Alexa 488 CTxB (Invitrogen) in the presence or not of 50 mU/ml SM. Cells were then washed twice in cold DME + 20% FBS, twice in cold acid buffer (0.2 M acetic acid, 0.5 M NaCl, pH 2.8) to remove extracellularly bound CTxB, followed by two additional washes in cold PBS. Cells were then fixed with 4% PFA, DAPI stained and mounted on slides before imaging with a Leica SPX5 confocal system with a 63 × N.A. 1.4 oil objective. Z stacks (0.13 μm Z step between optical sections) were acquired on a minimum of 5–10 random fields containing 32–66. Stacks of individual channels were then imported to Volocity Suite (PerkinElmer), the total fluorescence intensity of the channel per microscopic field was determined (Intensity × Voxel count), and the values were normalized by the number of cells in each field (determined by DAPI staining).

## Transferrin uptake and quantification

Cells treated with control or Dyn2 siRNA or induced or not with tamoxifen were incubated for 10 min with 4 μg/ml of Texas Red transferrin in FBS-free DME at 37°C. Cells were then washed twice in

cold PBS and twice in cold acid buffer containing 0.2 M acetic acid + 0.5 M NaCl, pH 2.8, followed by two more washes in cold PBS before fixation in 4% PFA and DAPI staining. Coverslips were imaged and the fluorescence intensity quantified as described for cholera toxin B-A488 uptake assays.

## Mechanical wounding

NRK cells treated with control or Cav1 siRNA (Invitrogen) were cultured to 70% confluence on 3 cm dishes with coverslips, in MatTek glassbottom dishes or 10 cm dishes, and sprinkled with 0.05 g (coverslips and Mattek dishes) or 0.1 g (10 cm dishes) ≤106 µm acid washed glass beads (Sigma) in DME ± $Ca^{2+}$, followed by gentle rocking as described in *Reddy et al. (2001)* and processed for PI staining, cavin pulldown or TEM as described above.

## PI exclusion PM repair assay

NRK cells or myotubes cultured at 70% confluence on six well dishes or *flexor digitorum brevis* mouse muscle fibers were treated with 250–800 ng/ml SLO for 5 min at 4°C or sprinkled with glass beads and incubated for 4 min in FBS-free DME ± $Ca^{2+}$ at 37°C, stained for 1 min with 50 µg/ml PI (Sigma), fixed in 4% PFA, DAPI stained and imaged immediately using an Axiovert 200 (Carl Zeiss, Inc., Jena, Germany) equipped with a CoolSNAP HQ camera (Roper Scientific) and MetaMorph software (MDS Analytical Technologies, Sunnyvale, CA). Quantifications were done by counting all nuclei stained with DAPI and PI in five random fields in triplicate (images taken with a 10 × or 32 × objective) and determining the percentage of PI positive nuclei. To induce large mechanical wounds, NRK cells treated with control or Cav1 siRNA for 48 hr were scraped from the dish in the presence or absence of $Ca^{2+}$, stained with PI after 4 min (for 5 min at 37°C) and analyzed by flow cytometry (at least 10,000 cells per sample) as described (*Tam et al., 2010*) (*Idone et al., 2008*). The data were analyzed using Flowjo software (Tree Star, Inc.).

## Lysosome exocytosis assays

Live immunofluorescence of surface-exposed Lamp1 was performed as previously described (*Reddy et al., 2001*) in differentiated C2C12 myotubes treated with 400 ng/ml of SLO for 5 min at 4°C and exposed to DME ± $Ca^{2+}$ for 15 min at 37°C before being washed with ice cold PBS and incubated with anti-mouse Lamp1 (1D4B) antibodies for 30 min at 4°C. Cells were then washed and fixed in 4% PFA, DAPI stained, incubated with anti-mouse Alexa Fluor 488 and imaged with an Axiovert 200 microscope as described above. Exocytosis of ASM by C2C12 myoblasts and myotubes was assessed by assaying enzyme activity in the supernatant of cultures treated with 400 ng/ml SLO ± $Ca^{2+}$ for various time points, as previously described (*Tam et al., 2010*).

## Dextran endocytosis assay

To visualize endosomes induced by SLO permeabilization in the presence or absence of $Ca^{2+}$, C2C12 myotubes were incubated or not with 400 ng/ml SLO for 5 min at 4°C and incubated in DME ± $Ca^{2+}$ at 37°C, in the presence of 2.5 mg/ml lysine-fixable 10 kDa Texas Red dextran (Invitrogen) for 4 min before fixation in 4% PFA and DAPI staining. Cells were then imaged in an Axiovert 200 microscope as described above.

## Immunofluorescence and co-localization analysis in myotubes

C2C12 myotubes cultured on coverslips were treated or not with 50 mU/ml SM for 30 s before fixation, quenched with 50 mM ammonium chloride, blocked with 5% FBS and permeabilized with 0.2% saponin in PBS. Cells were then incubated with rabbit anti-PTRF/Cavin1 and mouse anti-Cav3, 1:200 and 1:100 dilution in PBS 0.2% saponin respectively for 1 hr, followed by 1 hr incubation with secondary antibody conjugated with Alexa fluor 488 (anti-mouse) or Texas Red (anti-rabbit). Nuclei were stained with 10 µM DAPI. C2C12 cells differentiated for 5 days were immunolabeled for Cav3 and DAPI-stained to distinguish myoblasts (Cav3-negative) from myotubes (Cav3-positive). Images were acquired using a confocal Leica SPX5 microscope with a 63 × 1.4 N.A. oil objective. For each condition Z stacks (0.13 µm Z steps) were obtained from upper optical sections of the myotubes, and 12 fields were imaged for each condition. Colocalization coefficients were calculated using Volocity Suite (PerkinElmer). Each myotube (identified by Cav3 expression) was outlined by hand and defined as a region of interest. A total of 61 and 70 cells for control and SM, respectively, were analyzed.

## Acknowledgements

We thank EH Schuchman (Mount Sinai Medical Center) for recombinant ASM, R Tweten (U Oklahoma) for SLO expression plasmids and useful discussions, Jan F Endlich (University of Maryland) for EM assays, B Mittra and J Jensen (University of Maryland) for SLO expression and purification, S Ferguson and P DeCamilli (Yale University) for the inducible dynamin KO cell line, B Kachar (NIDCD, NIH) for support with image analysis, M Stober and A Helenius (ETH, Zurich) for advice and anti-EHD2 antibodies, A Upadhyaya (University of Maryland) for technical advice with TIRF microscopy, A Beaven (CBMG Imaging Core, University of Maryland) for assistance with confocal microscopy, M Graham and K Zichichi (Center for Cell and Molecular Imaging, Yale University School of Medicine) for cryo-immuno EM, and members of the Andrews laboratory for helpful discussions.

## Additional information

### Funding

| Funder | Grant reference number | Author |
| --- | --- | --- |
| National Institutes of Health | R01 GM064625 | Norma W Andrews |
| National Institutes of Health | R37 AI34867 | Norma W Andrews |
| National Council for Scientific and Technological Developement - Ciencia sem Fronteiras Program, Brazil | 200890/2012-3 | Thiago Castro-Gomes |
| CAPES Foundation, Ministry of Education of Brazil | | Patricia E Almeida |

The funders had no role in study design, data collection and interpretation, or the decision to submit the work for publication.

### Author contributions

MC, PEA, TC-G, Conception and design, Acquisition of data, Analysis and interpretation of data, Drafting or revising the article; CT, MCF, MC, HM, TKM, Acquisition of data, Analysis and interpretation of data; BAM, NWA, Conception and design, Analysis and interpretation of data, Drafting or revising the article; WS, Conception and design, Analysis and interpretation of data

### Ethics

Animal experimentation: This study was performed in strict accordance with the recommendations in the Guide for the Care and Use of Laboratory Animals of the National Institutes of Health. All of the animals were handled according to a protocol approved by the institutional animal care and use committee (IACUC) (#R-11-73) of the University of Maryland, College Park. All mice were sacrificed by $CO_2$ asfixiation, a euthanasia method that is fully consistent with the recommendations of the Panel on Euthanasia of the American Veterinary Medical Association and strictly recommended by the Central Animal Resources Facility of the University of Maryland. Every effort was made to minimize suffering.

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
