## [Decision Letter]

Thank you for sending your work entitled “Sphingomyelinase-dependent internalization of caveolae reseals wounded cells and muscle fibers” for consideration at *eLife*. Your article has been favorably evaluated by a Senior editor and 3 reviewers, one of whom is a member of our Board of Reviewing Editors.

The Reviewing editor and the other reviewers discussed their comments before we reached this decision, and the Reviewing editor has assembled the following comments to help you prepare a revised submission. The main point that the reviewers would like you to address involves showing that SLO is internalized in caveolar vesicles, not in communication with the extracellular space. Specifically:

1) The authors should do a direct endocytosis assay with SLO and ask if caveolin 1 is required for this. This should be easy enough given that they are set up for caveolin 1 siRNA experiments.

2) The authors should show that the caveolin-1-RFP construct that they use is a good reporter for the distribution of endogenous caveolin 1. This is very easy to do.

3) The authors should show that the very nice co-localisation between caveolin and SLO reported in Figure 1 represents *internalised* SLO, not SLO that is in caveolae still connected to the plasma membrane. This is maybe not so easy to do, but is certainly a very important thing to establish rigorously as it is the key part of the authors' model.

---

## [Author Response]

*1) The authors should do a direct endocytosis assay with SLO and ask if caveolin 1 is required for this. This should be easy enough given that they are set up for caveolin 1 siRNA experiments*.

We developed a direct SLO internalization assay using SLO labeled with Alexa 488 at position 66, in combination with a powerful anti-Alexa 488 quenching antibody. It is important to note that this was a challenging assay to set up because we could not use more extensive methods of amine-based labeling that would have resulted in brighter toxin – such methods are known to interfere with the oligomerization and extensive C-terminal conformational change that is required for SLO pore formation. Importantly, residues at the C-terminus of SLO are inserted in the lipid bilayer during pore formation, and in the process become protected from quenching. This was elegantly demonstrated by Rod Tweten, an expert in pore-formation by cholesterol-dependent toxins, who advised us in these experiments. He kindly provided us with a single cysteine SLO construct that can be labeled with thiol-reactive maleimide-Alexa 488 at the N-terminus, a region of the toxin that does not participate in oligomerization and membrane insertion, remaining exposed extracellularly after pore formation. Alexa 488 was chosen for the labeling because it is stable over a wide pH range, brighter than other fluorophores and very efficiently quenched by anti-Alexa 488 commercial antibodies.

The new data in Figure 2 shows that >90% of the labeled toxin was quenched at 4°C, demonstrating that in the absence of endocytosis any SLO bound to cells, including potentially to the lumen of caveolae (given the role of cholesterol as the SLO receptor) remain fully accessible to quenching antibodies. In contrast, at 37°C a much larger fraction of the cell-associated SLO was protected from quenching, as would be expected from a complete internalization process. The clear results of this assay argue strongly against the suggestion that the Cav1-positive caveolae containing SLO that we observed by cryo-immuno EM were still connected to the PM. The remaining quenchable SLO associated with cells that resealed their PM at 37°C is likely to correspond to bound but not fully oligomerized toxin.

Unfortunately, it is not technically possible to perform this quench-based assay in cells silenced for Cav1, because of their strong plasma membrane repair defect after SLO or mechanical wounding (as shown extensively in this manuscript with several cell types). The appropriate quenching of extracellular fluorescence requires an intact plasma membrane (SLO forms pores large enough to allow entry of the quenching antibodies). Thus, as an alternative approach to complement our cryo-immuno EM colocalization of SLO and Cav1, we performed confocal microscopy in cells incubated with Alexa 488-SLO and treated with quenching anti-Alexa 488 antibodies. The quenching antibody activity was evident in these images, and some of the Alexa 488-SLO that survived quenching was observed associated with Cav1-positive intracellular pucta (new Figure 3).

*2) The authors should show that the caveolin-1-RFP construct that they use is a good reporter for the distribution of endogenous caveolin 1. This is very easy to do*.

The new Figure 4 shows that after expression in HeLa cells mRFP-Cav1 traffics correctly to punctate structures containing endogenous Cav1.

*3) The authors should show that the very nice co-localization between caveolin and SLO reported in*
Figure 1
*represents* internalised *SLO, not SLO that is in caveolae still connected to the plasma membrane. This is maybe not so easy to do, but is certainly a very important thing to establish rigorously as it is the key part of the authors' model*.

We agree that this was an important thing to do. As explained above in our response to point 1, the new FACS-based endocytosis assay demonstrates that Alexa 488-SLO enters compartments that pinch off from the PM, and the new confocal images show that labeled toxin protected from the quenching antibodies colocalizes intracellularly with Cav1. We also tried to repeat the cryo-immuno EM assays using ruthenium red and other membrane impermeable dyes, but these agents destroyed protein antigenicity.